# Zyxin contributes to coupling between cell junctions and contractile actomyosin networks during apical constriction

Mark M. Slabodnick[1,2,☯,*], Sophia C. Tintori[1,☯,¤], Mangal Prakash[3], Pu Zhang[1], Christopher D. Higgins[1], Alicia H. Chen[1], Timothy D. Cupp[1], Terrence Wong[1], Emily Bowie[1], Florian Jug[3,4], Bob Goldstein[1,5]

**1** Biology Department, University of North Carolina at Chapel Hill, Chapel Hill, North Carolina, United States of America, **2** Department of Biology, Knox College, Galesburg, Illinois, United States of America, **3** Max Planck Institute of Molecular Cell Biology and Genetics, Dresden, Germany, **4** Fondazione Human Technopole, Milan, Italy, **5** Lineberger Comprehensive Cancer Center, University of North Carolina at Chapel Hill, Chapel Hill, North Carolina, United States of America

☯ These authors contributed equally to this work.
¤ Current address: Department of Biology, and Center for Genomics and Systems Biology, New York University, New York, New York, United States of America
* mmslabodnick@knox.edu

**Data Availability Statement:** C. elegans strains generated in this study have been deposited to the Caenorhabditis Genetics Center, accession numbers can be found in S1 Table. An interactive

## Abstract

One of the most common cell shape changes driving morphogenesis in diverse animals is the constriction of the apical cell surface. Apical constriction depends on contraction of an actomyosin network in the apical cell cortex, but such actomyosin networks have been shown to undergo continual, conveyor belt-like contractions before the shrinking of an apical surface begins. This finding suggests that apical constriction is not necessarily triggered by the contraction of actomyosin networks, but rather can be triggered by unidentified, temporally-regulated mechanical links between actomyosin and junctions. Here, we used *C. elegans* gastrulation as a model to seek genes that contribute to such dynamic linkage. We found that α-catenin and β-catenin initially failed to move centripetally with contracting cortical actomyosin networks, suggesting that linkage is regulated between intact cadherin-catenin complexes and actomyosin. We used proteomic and transcriptomic approaches to identify new players, including the candidate linkers AFD-1/afadin and ZYX-1/zyxin, as contributing to *C. elegans* gastrulation. We found that ZYX-1/zyxin is among a family of LIM domain proteins that have transcripts that become enriched in multiple cells just before they undergo apical constriction. We developed a semi-automated image analysis tool and used it to find that ZYX-1/zyxin contributes to cell-cell junctions' centripetal movement in concert with contracting actomyosin networks. These results identify several new genes that contribute to *C. elegans* gastrulation, and they identify zyxin as a key protein important for actomyosin networks to effectively pull cell-cell junctions inward during apical constriction. The transcriptional upregulation of ZYX-1/zyxin in specific cells in *C. elegans* points to one way that developmental patterning spatiotemporally regulates cell biological mechanisms *in vivo*. Because zyxin and related proteins contribute to membrane-cytoskeleton linkage in

online visualization of transcriptomic data is available at https://n2t.net/ark:/84478/d/2bbpmsq3. RNA-seq reads, alignments, and RPKM files are available on NCBI GEO accession number GSE205061. A Jupyter Notebook with the code for the image analysis is available on GitHub (https://github.com/fjug/BobSeg). All other relevant data are in the manuscript and its Supporting information files.

**Funding:** This work was supported by NIH (https://www.nih.gov/grants-funding) R35GM134838 to BG, NSF (https://beta.nsf.gov/funding) GRFP, NIH (https://www.nih.gov/grants-funding) F31 HD088128 and Damon Runyan Fellowship (https://www.damonrunyon.org/for-scientists/award programs) DRG-2371-19 to SCT and NIH (https://www.nih.gov/grants-funding) F32GM119348 to MMS. The funders had no role in study design, data collection and analysis, decision to publish, or preparation of the manuscript.

**Competing interests:** The authors have declared that no competing interests exist.

other systems, we anticipate that its roles in regulating apical constriction in this manner may be conserved.

## Author summary

Animals take shape during development in large part by the bending of tissues. Failures in this process are common causes of human birth defects. Such tissue bending is driven primarily by individual cells changing shape: in many examples, one side of a cell shrinks, pulling on junctions that connect the cell to neighboring cells. But the networks that drive one side of a cell to shrink are not always connected to junctions. As a result, focus has turned to understanding how connections between such networks and junctions are dynamically regulated to trigger cell shape change. We sought to identify genes that contribute to these dynamic connections. Here, we describe proteomic and transcriptomic methods that we used to identify proteins that contribute to cell shape change. We developed a new image analysis tool and used it to reveal that loss of one of these genes results in networks moving without efficiently pulling in junctions. Our results identify new molecular players, and they pinpoint a key gene whose products might contribute to dynamically connecting networks to junctions to trigger tissue shape changes in *C. elegans* and other organisms.

## Introduction

During embryogenesis, molecular forces drive the tissue shape changes that give form to the developing organism [1]. Among the mechanisms that drive such tissue shape changes, apical constriction is one of the most commonly used [2]. For example, the neural tube of vertebrate embryos forms as some cells of the neural plate constrict apically, bending the neural plate into a tube and internalizing from the embryo's surface [3]. Neural tube formation fails frequently in human development [4]. Understanding the mechanisms that control changes to cell shape is essential to understanding disease states as well as the fundamental mechanisms by which embryos develop.

The force-producing mechanisms that drive apical constriction are well conserved, relying on cortical networks of actin filaments and non-muscle myosin II motors, which drive contraction of the apical cell cortex [5]. The forces that contract the apical cell cortex are transmitted to neighboring cells through apical cell junctions. As a result, the contraction of a cortical network can shrink the exposed apical surface of the cell [6,7]. How this mechanism is developmentally regulated, driving specific cells to constrict only their apical surfaces and at specific times, remains incompletely understood in most model systems [2,8].

The nematode *Caenorhabditis elegans* has been a valuable model for studying mechanisms of morphogenesis [9]. Gastrulation in *C. elegans* begins at the 26-cell stage when a non-muscle myosin II becomes enriched in the apical cortex of two endodermal precursor cells (EPCs) [10], which then internalize by apical constriction [11,12].

A previous study investigated actomyosin dynamics in the EPCs during gastrulation and found, unexpectedly, that contractions of the actomyosin cortex initially occurred in a conveyer belt-like fashion without pulling junctions centripetally, i.e., with junctions apparently uncoupled to the inward movement of actomyosin components toward the center of the apical cell surface [13]. It was only after several minutes of seemingly unproductive actomyosin

contraction that cell-cell junctions began to move increasingly in concert with the contracting networks. This phenomenon was also observed in *Drosophila melanogaster* shortly before ventral furrow formation, where myosin accumulated and coalesced periodically in weak contractions that preceded the shrinking of apical cell profiles [13]. These observations suggest that in these model systems, and potentially more generally, apical constriction is not triggered by myosin activation; rather, it is likely to be triggered by gradually connecting an already-contracting apical actomyosin cytoskeleton to cell-cell junctions, via unknown links.

Although many proteins have been identified at sites of cadherin-based adhesion that could feasibly serve as such temporally regulated links [14], the specific, temporally-regulated links relevant to triggering apical constriction are not yet known in any system. Identifying such temporally-regulated links from among candidate linkers or unknown players is an important step toward understanding how developmental mechanisms orchestrate the cytoskeletal mechanisms of apical constriction with spatial and temporal precision.

Here, we sought to identify proteins that could contribute to such temporally-regulated linkage either directly or indirectly. We anticipated that identifying key proteins would require integrating diverse methodologies including generating new image analysis resources, as well as screening for defects in a complex process involving cellular and subcellular dynamics. First, we hypothesized that temporal regulation of apical constriction could feasibly result from either the transcriptional regulation or posttranscriptional regulation of key linking proteins, or both. We found that members of the *C. elegans* cadherin-catenin complex (CCC) remained at apical cell-cell contacts as cortical actomyosin contractions began, suggesting that disassembly of these complexes cannot explain junctions' initial failure to move. We then used both proteomic and transcriptomic approaches to find proteins that might interact physically with the *C. elegans* CCC as well as genes whose expression is upregulated specifically in the EPCs prior to cell internalization. Screening through the resulting list identified several new contributors to *C. elegans* gastrulation, including two genes, *afd-1*/afadin and *zyx-1*/zyxin, that encode candidate linkers and that we found were required for timely internalization of the EPCs. We found that AFD-1/afadin is localized broadly to cell junctions with patterns consistent with members of the CCC. Single-cell RNA-seq on multiple internalizing cells identified *zyx-1*/zyxin and other LIM domain-encoding genes as upregulated in multiple internalizing *C. elegans* cell lineages. To determine whether *zyx-1*/zyxin and *afd-1*/afadin contribute to linking contracting actomyosin networks to junctions *in vivo*, we developed a new, semi-automated image analysis workflow for quantifying actomyosin network and junctional movements. The results identified zyxin as a contributor to triggering apical constriction through regulating directly or indirectly the dynamic, temporally-regulated coupling of actomyosin contractions to cell-cell junctions.

## Results

### Cortical actomyosin initially contracts away from stably positioned cadherin and catenins

We considered that the failure of initial contractions of actomyosin networks to pull cell-cell junctions inward in *C. elegans* gastrulation [13] might be explained by an initial failure of α-catenin and/or β-catenin to remain stably at cell-cell junctions with cadherin. To test this hypothesis, we filmed embryos expressing fluorescent tags inserted into the native locus of each protein without disrupting each protein's function [15]. As expected, mKate2-tagged HMR-1/cadherin localized at apical cell-cell boundaries, as did GFP::HMP-2/β-catenin and HMP-1/α-catenin::GFP (Fig 1A and 1B). We visualized the movements of actomyosin along with the CCC components during the early stage, when there are uncoupled contractions (2–8 minutes after the initiation of cytokinetic furrow formation in MSa and MSp cells [13]) using

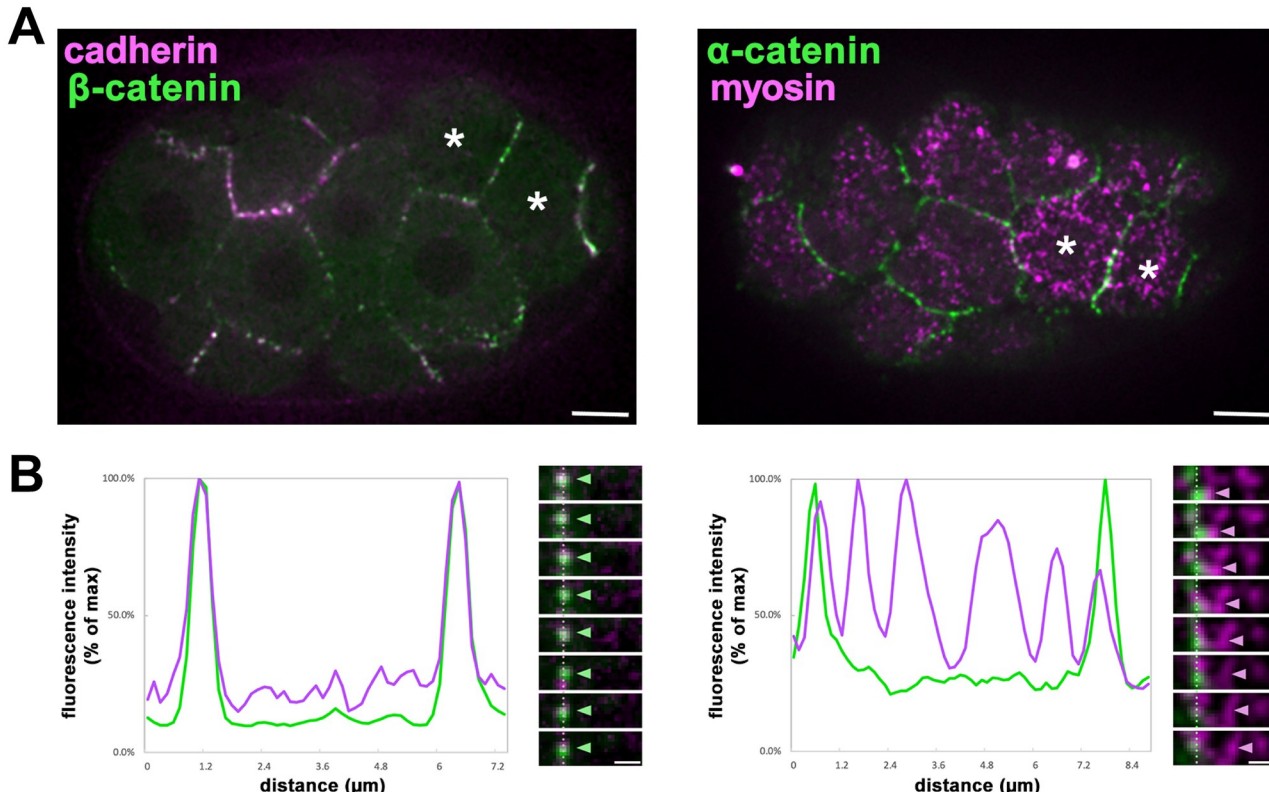

**Fig 1. All three CCC components remain associated with apical membrane borders during early stage actomyosin contractions. (A)** Two-color spinning disk confocal fluorescence images of HMR-1::mKate2 (cadherin) with GFP::HMP-2 (β-catenin), left, and HMP-1::GFP (α-catenin) with NMY-2::mKate2 (myosin), right. Apically constricting cells are labeled (asterisks) **(B)** Representative line scans of the fluorescence intensities across the cell cortex from anterior (left) to posterior (right) with corresponding images of these junctions over time. As indicated by arrowheads, β-catenin (n = 3) and α-catenin (n = 7) remained at apical membrane borders while myosin particles moved centripetally. (A) Scale Bar = 5 μm (B) inset scale bar = 1 μm, kymograph frames represent 3 second intervals.

strains co-expressing a red fluorescently tagged non-muscle myosin II heavy chain (NMY-2:: mKate) along with a GFP-tagged CCC component. We mounted these embryos ventrally to visualize *en face* centripetal actomyosin dynamics in the apical cortices of endoderm precursor cells (EPCs). During this early stage, we observed robust centripetal movement of myosin particles, but the bulk of the α-catenin-GFP and GFP-β-catenin remained stably at apical cell-cell boundaries, and failed to move centripetally (Fig 1B). We conclude that the initial inability of contracting actomyosin networks to efficiently pull cell-cell junctions centripetally cannot be explained by α-catenin and/or β-catenin being pulled away from cell-cell junctions. Although F-actin can associate with α-catenin various organisms including *C. elegans* [16,17], and this connection can be strengthened under force in some systems [18], these results imply that a strong connection between the contracting apical actomyosin network and junctional α-catenin is initially missing in this system (Fig 1B). We next sought to identify proteins that might contribute to this connection.

## Identification of candidate proteins that might contribute to coupling of contracting actomyosin networks and junctions

We used two screening approaches to identify proteins that could feasibly contribute to connecting actomyosin to junctions in EPCs: We screened for proteins from early-stage

embryos that co-immunoprecipitate (co-IP) with α-catenin, and proteins whose mRNAs became enriched in EPCs just prior to the onset of apical constriction. To identify proteins that co-IP with α-catenin, we used a strain expressing endogenously-tagged HMP-1::GFP/α-catenin. We performed co-IP using anti-GFP antibodies to pull down the CCC and any associated proteins. Because we were interested in proteins present during gastrulation in early embryogenesis, we enriched our samples for early stage (<50 cell) embryos (see Materials and methods). We used a strain expressing soluble GFP alone as a control from which to subtract contaminating proteins that co-purify with GFP. Our initial list of co-IP'd proteins contained each of the other CCC proteins at high peptide counts as expected, confirming that we pulled down intact CCCs from early stage embryos (S3 Table). Our list also contained more than 200 other proteins with at least one detectable peptide. We expect that this list includes proteins that interact with α-catenin in one or more cells as well as false positives. We view the possibility that the list may be enriched for gene products of interest as sufficient for our purpose of further screening. We further narrowed this list to only candidates whose genes were predicted to be expressed before or during the 24-cell stage of development using published single-cell mRNA sequencing data [19], and we removed common housekeeping genes (see Materials and methods). This resulted in 11 candidates with the potential to physically interact with the CCC for further screening (Table 1).

Gastrulation in *C. elegans* relies on embryonic transcription [20]. Therefore it is possible that some proteins that contribute to the dynamically regulated linkage between actomyosin and junctions are encoded by genes that become transcribed in EPCs near the time that these cells are born. To identify a second set of candidates from genes expressed specifically in EPCs, we examined our previously-generated single-cell RNA-seq data of the first several cell cycles of development [19]. We considered genes whose mRNAs were enriched in EPCs compared to the rest of the embryo just prior to apical constriction (i.e. in the endodermal precursor cell E at the 8-cell stage, or its daughter cells Ea and Ep at the 24-cell stage) and enriched compared to cells of earlier stages. From this list, we selected 21 genes whose mRNAs were enriched at least 8-fold in EPCs vs the rest of the embryo at the 8-cell stage (16 genes) or the 24-cell stage (5 genes).

To identify genes from both lists above that contribute to gastrulation *in vivo*, we then screened candidates by RNA interference (RNAi). Rather than feeding bacteria expressing double-stranded RNAs (dsRNAs), we used the more laborious method of injecting dsRNAs targeting each gene in order to maximize the likelihood of strongly disrupting gene functions [21]. We filmed embryos released from injected mothers and examined them for a gastrulation-defective (Gad) phenotype, defined as the two EPCs failing to fully internalize (i.e. with part of at least one of the cells not covered by any other cells) before dividing. Our RNAi screening identified 21 candidate genes with at least a low frequency Gad phenotype. These results define a set of genes that will be of interest for future studies of gastrulation mechanisms (Table 1).

Among the genes we identified were two we chose to focus on because they encode proteins that associate directly or indirectly with junctional proteins and/or actin networks in *C. elegans* or other organisms: *afd-1*/afadin and *zyx-1*/zyxin [14,22–25]. The *Drosophila* afadin homolog Canoe is required along with β-catenin for medioapical actomyosin to remain connected to adherens junctions during apical constriction [26,27], although it has not been implicated in the kind of temporally-regulated linkage during normal development that we sought here. We first pursued *afd-1*/afadin's roles in *C. elegans* apical constriction. *zyx-1*/zyxin is discussed further below.

**Table 1. Genes Identified from a Combined Proteomic and Transcriptomic Screen for candidates.** Gene names, α-catenin co-IP peptide counts, and EPC mRNA enrichment for each candidate are presented. Enrichment data are shown for genes with EPC enrichment over the whole embryo at that stage if EPC(s) also had higher RPKM than cells at earlier stages. Primers were designed to amplify each target gene sequence from cDNA (S2 Table). Each dsRNA was injected into young adults of the wild-type *N2* strain of *C. elegans*. Embryos laid by injected worms were scored 24 hours after injection, filmed by DIC microscopy, and examined for Gad phenotypes. *gdi-1* RNAi resulted in 100% sterility, failing to produce embryos for analysis, as seen previously [28].

| Screen | dsRNA target | Peptide Counts | Stage Enriched (Cell #) | Enrichment (Mean RPKM in EPC over Embryo) | % Gad (#/N) |
|---|---|---|---|---|---|
| N/A | Negative Control | N/A | N/A | - - | 0 (0/20) |
| co-IP | C01H6.2 / *mlt-2* | 5 | 24 | 40 / 7 | 21.4 (3/14) |
| co-IP | *afd-1* | 1 | N/A | Not Enriched | 25 (5/20) |
| co-IP | F44E5.1 | 3 | N/A | Not Enriched | 5.8 (1/17) |
| co-IP | *gdi-1* | 3 | N/A | Not Enriched | 100% sterile |
| co-IP | *gei-4* | 1 | N/A | Not Enriched | 41.6 (5/12) |
| co-IP | *inx-3* | 1 | N/A | Not Enriched | 0 (0/12) |
| co-IP | *lin-66* | 2 | N/A | Not Enriched | 0 (0/9) |
| co-IP | *noah-1* | 2 | 24 | 71 / 9 | 30.8 (4/13) |
| co-IP | *sym-1* | 2 | 24 | 6 / 2 | 0 (0/17) |
| co-IP | T19B10.2 | 2 | 24 | 416 / 46 | 0 (0/11) |
| co-IP | Y38H6C.14 | 1 | 8; 24 | 15 / 3; 21 / 4 | 29.4 (5/17) |
| EPC Enrichment | *acp-2* | 0 | 24 | 1084 / 121 | 12.5 (2/16) |
| EPC Enrichment | *add-1* | 0 | 8 | 11 / 0 | 7.7 (1/13) |
| EPC Enrichment | C26F1.1 | 0 | 24 | 879 / 98 | 0 (0/9) |
| EPC Enrichment | C29F7.2 | 0 | 24 | 1231 / 137 | 27.3 (3/11) |
| EPC Enrichment | C46E10.8 | 0 | 8 | 146 / 20 | 50.0 (6/12) |
| EPC Enrichment | *ctn-1* | 0 | 8 | 65 / 14 | 20.0 (2/10) |
| EPC Enrichment | *dve-1* | 0 | 24 | 1124 / 126 | 0 (0/16) |
| EPC Enrichment | F25D7.5 | 0 | 8 | 52 / 7 | 0 (0/15) |
| EPC Enrichment | F49E10.4 | 0 | 8 | 171 / 22 | 0 (0/13) |
| EPC Enrichment | *grdn-1* | 0 | 8 | 13 / 3 | 33.3 (3/9) |
| EPC Enrichment | H24G06.1 | 0 | 8 | 64 / 9 | 14.3 (2/14) |
| EPC Enrichment | *hum-8* | 0 | 24 | 33 / 4 | 66.7 (6/9) |
| EPC Enrichment | *let-4* | 0 | 8 | 34 / 4 | 0 (0/17) |
| EPC Enrichment | *pssy-1* | 0 | 8 | 179 / 38 | 7.1 (1/14) |
| EPC Enrichment | R06B10.2 | 0 | 8 | 8 / 1 | 71.4 (5/7) |
| EPC Enrichment | T14E8.1 | 0 | 8 | 18 / 3 | 18.2 (2/12) |
| EPC Enrichment | *tnc-2* | 0 | 8 | 68 / 9 | 0 (0/9) |
| EPC Enrichment | Y53C10A.10 | 0 | 8 | 3 / 0 | 8.3 (2/12) |
| EPC Enrichment | Y57G11C.6 | 0 | 8 | 41 / 5 | 30 (3/10) |
| EPC Enrichment | *zig-5* | 0 | 8 | 34 / 5 | 31.3 (5/16) |
| EPC Enrichment | *zyx-1* | 0 | 8 | 96 / 12 | 25 (5/20) |

### *afd-1*/afadin contributes to gastrulation and co-localizes with the CCC

Afadin is an actin-binding protein and a broadly-conserved component of adherens junctions that localizes to sites under tension [22,27,29–33]. *C. elegans afd-1*/afadin genetically interacts with *sax-7*/L1CAM, which has functional redundancy with the CCC during gastrulation [30,34], and AFD-1/afadin had been found previously to co-IP with CCC components in *C. elegans* [35]. We found that *afd-1* RNAi resulted in Gad phenotypes in 25% of embryos (Fig 2A), and in all cases these defects were subtle: Only a small portion of the apically constricting EPC surface remained exposed to the exterior at the time of the next cell division (Fig 2A and 2B), suggesting that AFD-1/afadin has a partially redundant role in *C. elegans* gastrulation.

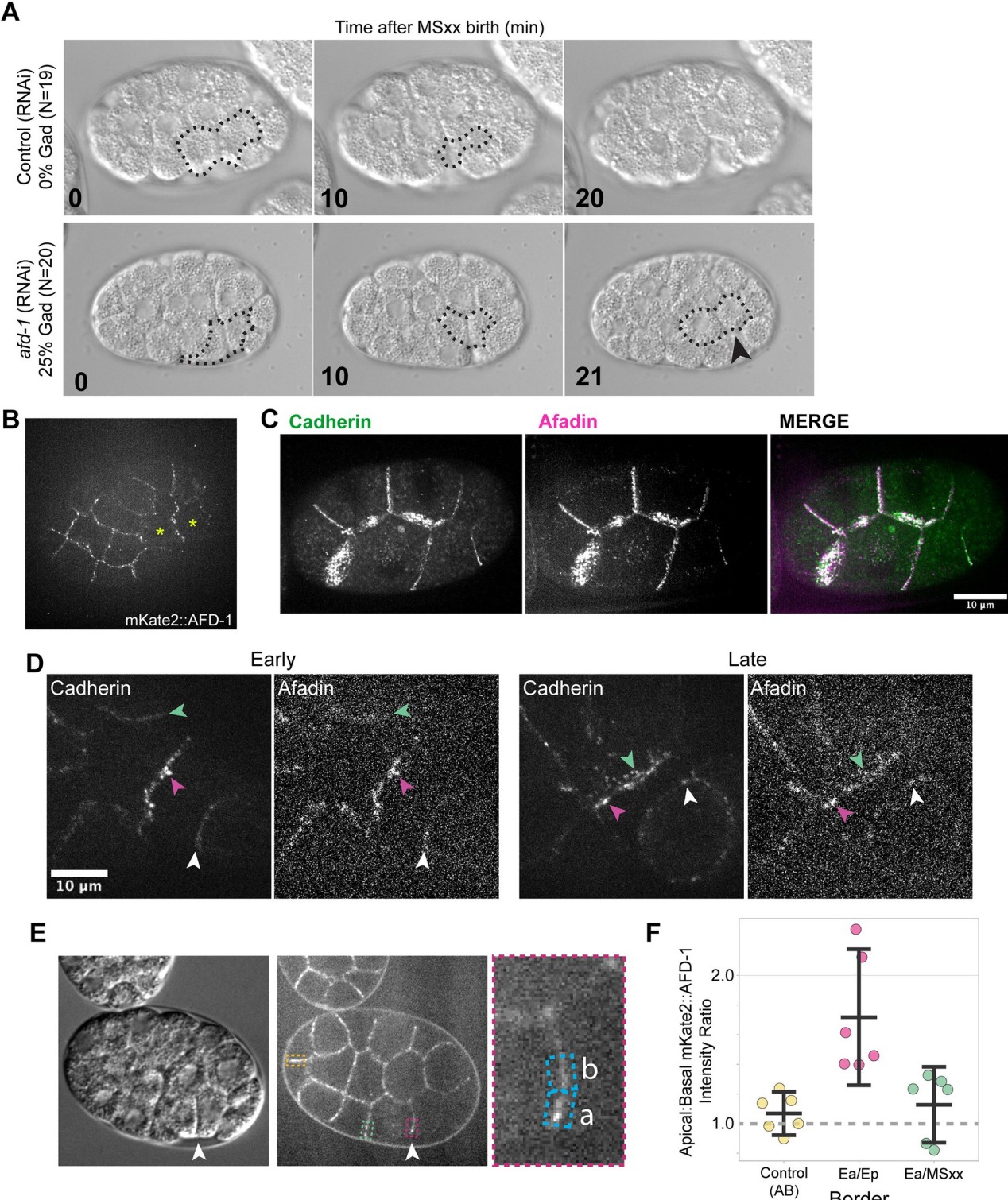

**Fig 2. Afadin contributes to gastrulation and localizes along with members of the CCC at cell junctions. (A)** DIC imaging of a near-lateral surface of *C. elegans* embryos around the start of gastrulation, from uninjected control mothers or those injected with dsRNA targeting *afd-1*. The surfaces of internalizing EPCs that were not covered by other cells in 3 dimensions are outlined (identified in multiplane videos; a single plane is shown), and the arrowhead points to these exposed EPC cell surfaces at the time of cell division, indicating a gastrulation defect that was not seen in the control embryos. **(B)** Single spinning-disk confocal section near the surface of a ventrally-mounted embryo showing junctional localization of endogenously expressed mKate2::AFD-1 protein. Apically constricting cells are labeled with yellow asterisks. **(C)** Max-intensity projection of a 8-cell stage embryo

with endogenously tagged HMR-1::GFP and mKate2::AFD-1 showing colocalization at junctions. **(D)** Representative max-intensity projections of HMR-1::GFP and mKate2::AFD-1 at both early and late time points of gastrulation. AFD-1 is present at junctions and colocalizes with HMR-1 at Ea/MSxx border (green arrowhead), Ea/Ep border (magenta arrowhead), and the Ep/P4 border (white arrowhead). **(E)** Single plane from a laterally mounted embryo showing that the border between apically constricting cells (arrowhead) has apically enriched mKate2::AFD-1 (magenta box), and enrichment that is less apically-biased at both the Ea/MSxx (green box) and AB control border (yellow box), quantified in (F). **(F)** Plot showing the ratio of Apical:Basal fluorescence intensities at the Ea/Ep border and control border. Mean values and 95% confidence intervals are shown, with dotted lines connecting pairs of measurements from each embryo (n = 6, *p* = 0.0002).

To gain more insight into AFD-1/afadin's role at junctions, we examined its localization by using CRISPR to endogenously tag the N-terminus of *afd-1* with the red fluorophore mKate2. mKate2::AFD-1 was present at cell-cell boundaries in general, including apical cell-cell boundaries, and it was apically enriched near the border between the two apically constricting EPCs, where CCC components are known to be enriched as well [15] (Fig 2B–2F). To determine if junctional mKate2:AFD-1 colocalized with members of the CCC, we generated a dual-labeled strain containing both mKate2::AFD-1 and a functional, endogenously-tagged HMR-1/cadherin::GFP [15]. In early embryos we found that the two proteins colocalized at cell junctions (Fig 2C) throughout the early and late stages of EPC internalization (Fig 2D, arrowheads).

Previously, interactions have been found between afadin and α-catenin in mammalian systems [36,37]. To determine if any CCC components influence mKate2::AFD-1's ability to localize to junctions, we targeted other CCC components by RNAi (Fig 3A and 3B). Interestingly, we saw a marked reduction of junctional levels of mKate2::AFD-1 in embryos injected with *hmr-1*/cadherin dsRNA but not *hmp-1*/α-catenin dsRNA, suggesting that in *C. elegans*, afadin requires HMR-1/cadherin but not HMP-1/α-catenin for recruitment to junctions (Fig 3C–3E). We similarly investigated a potential role for SAX-7/L1CAM in recruiting AFD-1 to junctions but did not see a reduction in mKate2::AFD-1 junctional levels in *sax-7* RNAi embryos (Fig 3C–3E). Taken together, we conclude that during gastrulation AFD-1/afadin localizes to adherens junctions, where it is recruited directly or indirectly by HMR-1/cadherin.

### RNA-seq of internalizing cell types to search for transcripts enriched in apically constricting cells of multiple cell lineages

Multiple cell lineages of the early *C. elegans* embryo internalize by apical constriction [10,38]. Our identification of 21 genes with enriched expression in just one of these lineages, the EPCs, made us wonder if there exist any *C. elegans* genes with expression enriched in multiple independently internalizing cell types. No such gene might exist, but we considered this issue worth investigating because if such a gene existed, we would view it as a candidate for orchestrating apical constriction, akin to snail family genes that orchestrate another cell shape change–epithelial-to-mesenchymal transitions–in multiple animal models [39–42]. Alternatively, finding that no such gene exists by exhaustive RNAseq analysis could also inform future models of apical constriction mechanisms by suggesting that apical constriction may be orchestrated by different regulators in different cells.

We collected cells from multiple internalizing cell lineages for RNA-seq (S1 Fig). We selected four groups of internalizing cells–MS descendants, E descendants, D descendants, and descendants of Cap and Cpp, hereafter referred to together as Cxp descendants (Fig 4A, red circles). We selected two non-internalizing groups as negative controls–ABp descendants (because only 2 out of 32 great-great granddaughter cells of ABp internalize) and Cxa descendants (none of which internalize) [38]. For each of the internalizing cell types, we sought transcripts whose abundance increased during the cell cycles leading up to internalization. To do

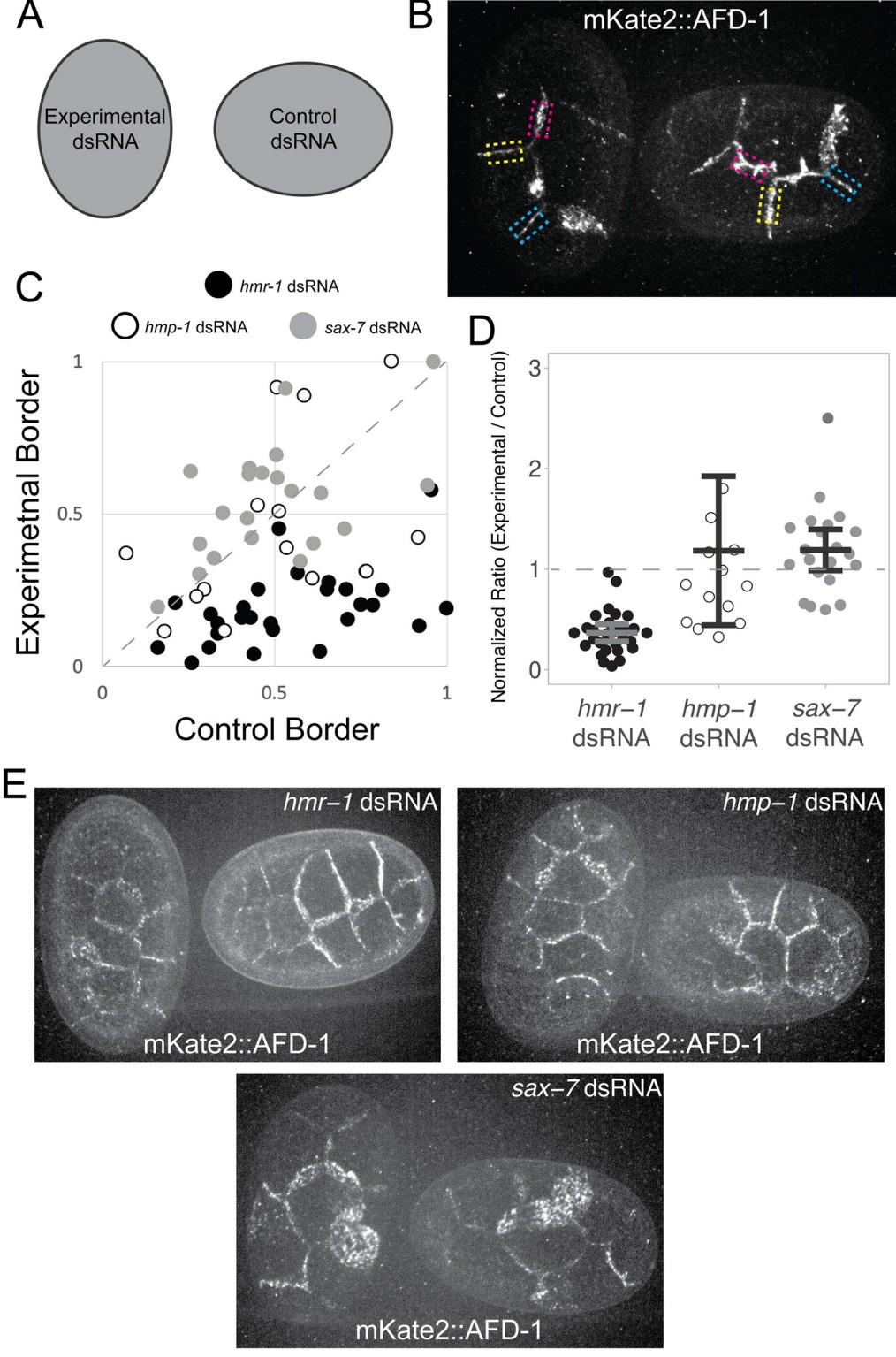

**Fig 3. AFD-1/afadin localization depends on *hmr-1* but not *hmp-1*.** **(A)** Embryos were mounted adjacent to one another oriented so that embryos from control and experimental dsRNA injected worms could be differentiated. **(B)** Embryos were paired by developmental stage, *hmr-1* RNAi vs. control RNAi shown here, and like junctions were analyzed between each pair of embryos (yellow, cyan, and magenta boxes indicate like junctions; yellow is an ABpl-ABal cell junction, magenta is an ABpl-MS cell junction, and blue is an ABal-ABar cell junction). **(C)** Plot of normalized

fluorescence intensities for control vs. experimental borders in *hmr-1* RNAi (filled circles), *hmp-1* RNAi (open circles), and *sax-7* RNAi (gray circles). The dotted line along the diagonal indicates a 1:1 ratio. **(D)** Data from C plotted as normalized ratios (experimental/control levels) with individual measurements, means and 95% confidence intervals shown. *hmr-1* RNAi embryos had a lower level of mKate2::AFD-1 at junctions than both *hmp-1* RNAi ($p = 0.0285$) and *sax-7* RNAi ($p = 1 \times 10^{-8}$). A single outlier data point with a value of 5.21 for the *hmp-1* dsRNA condition is not shown but was included in the statistical analysis. **(E)** Representative maximum projections of embryos quantified in **(C, D)** injected with the indicated dsRNA expressing mKate2::AFD-1.

this, we collected two transcriptomes from each internalizing cell lineage–one before internalization (2–3 cell cycles before cell internalization), and one at the start of internalization. The two non-internalizing cell types were used to exclude transcripts that became broadly enriched in all cell types at the relevant stages (Fig 4A, black circles). All transcriptomes from the 1- to 15-cell stages were previously published [19], and in this study we expanded the previous resource with transcriptomes from internalizing lineages. We present the results of the cumulative transcript dataset as a resource in an interactive online form, to facilitate querying the dataset, using the Differential Expression Gene Explorer, at http://dredge.bio.unc.edu/c-elegans-transcriptional-lineage-with-late-gastrulation/ (Permalink: https://n2t.net/ark:/84478/d/2bbpmsq3) [43].

To identify any transcripts that became enriched in internalizing cell types, we filtered the 7,998 transcripts that we detected (see Materials and methods) for those that became enriched at least two-fold over time in at least two of the four internalizing cell types, and that did not become enriched by more than two-fold in the two negative control, non-internalizing cell types. This analysis yielded 839 genes. Of the four internalizing cell types sampled, all but the E lineage generate some muscle cells (all D descendants, all Cxp descendants, and 17/52 MS descendants will become body muscle) [44]. To avoid genes that are exclusively associated with muscle fate, we filtered the 839 transcripts for those that became enriched over time in the E lineage and at least one other internalizing lineage, and then for only genes whose transcripts are enriched by at least two-fold in the internalizing C descendants (Cxp) compared to the non-internalizing C descendants (Cxa). This reduced our filtered list to 150 genes. From this list we removed genes whose maximum transcript abundance in any non-internalizing cell types exceeded by more than two-fold the transcript abundance in the internalizing cell types where enrichment had been found. Of 99 genes that remained (Fig 4B), 55 transcripts were enriched in two of the four internalizing cell types, 33 were enriched in three of the four, and 11 were enriched in all four (as in the complete internalization-correlated pattern shown in Fig 4C). We found that all 11 were either only weakly expressed in some of the internalizing cells or exhibited relatively high expression in some non-internalizing cells as well (S3 Fig). We conclude that no single gene can be found by this kind of RNA-seq analysis that satisfies our expectations for a *C. elegans* orchestrator of apical constriction in multiple cell lineages. Therefore, we considered next the possibility that multiple members of a gene family might together fulfill this expected pattern.

## Transcripts encoding a group of LIM domain-containing proteins become enriched in multiple apically constricting cells

To expand our analysis to include groups of genes that are similar to each other in sequence, we created groups of genes based on similarity (see Materials and methods), calculating a cumulative transcript profile for each such homology group by summing the transcript profiles of all the genes in the group (as shown in the last pictogram in Fig 4C). We evaluated each of these summed transcript profiles as we had the individual genes above. This analysis yielded

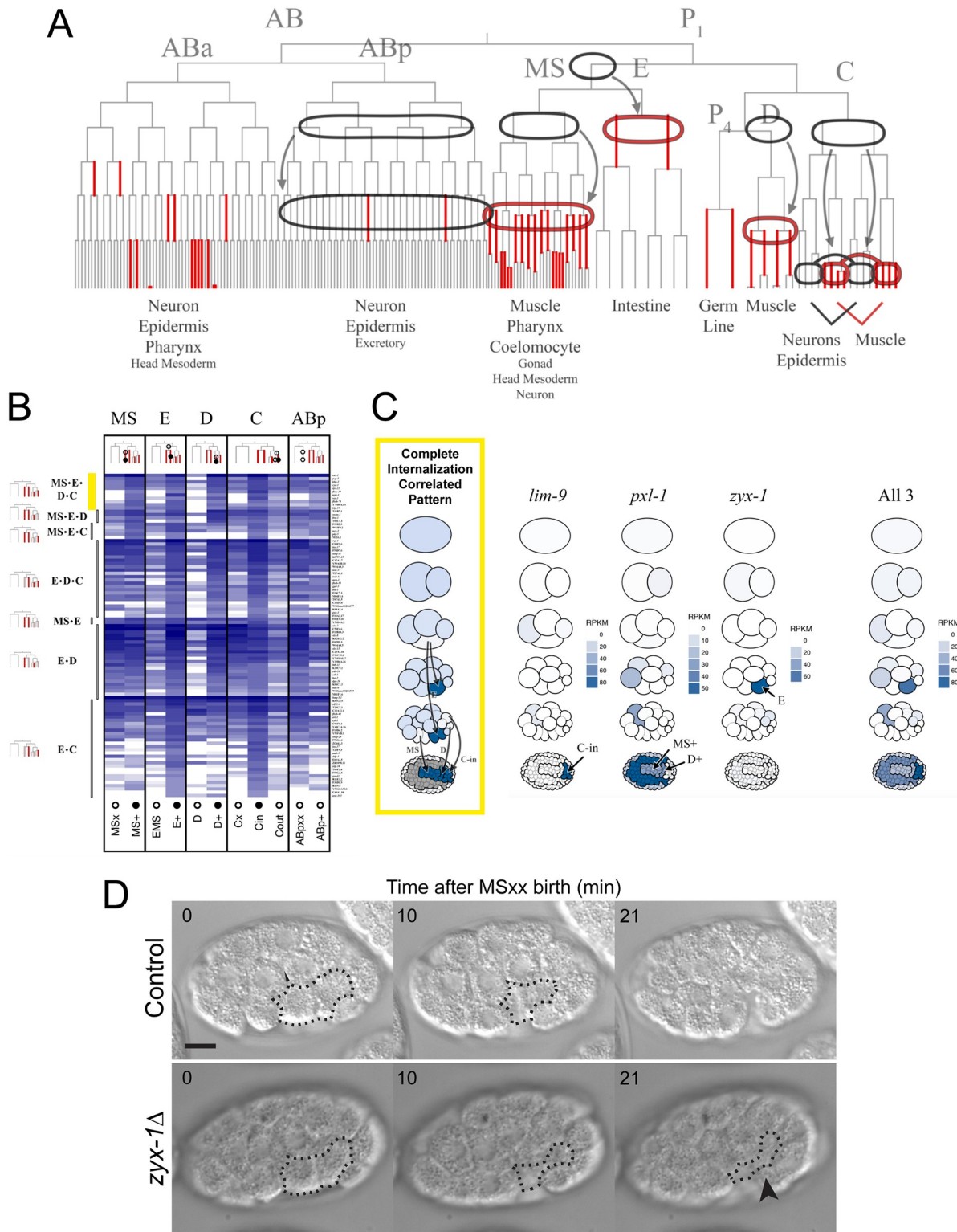

**Fig 4. Transcriptome profiling of multiple divergent, gastrulating cell types reveals internalization-correlated expression pattern of LIM domain-containing gene family.** (**A**) *C. elegans* lineage map indicating the four internalizing lineages. All internalizing cells are represented with red branches. The internalizing MS, E, D, and Cxp lineages were dissected for transcriptome profiling and compared against two negative control lineages that do not internalize—the ABp and Cxa lineages. (**B**) Heatmap showing transcript abundance for the 99 internalization-correlated genes. Open circles indicate samples from a non-internalizing cell while closed circles indicate samples from an internalizing cell.

**(C)** Pictograms of embryos representing relative gene expression levels of LIM domain-containing genes in the different internalizing cell lineages (key in S3 Fig). A mockup of an idealized gene expression pattern for a hypothesized orchestrator of gastrulation is shown on the left. The combined gene expression of all three LIM domain-containing genes is shown on the right. **(D)** DIC imaging of *C. elegans* embryos around the start of gastrulation in control (N2) embryos and in *zyx-1Δ* (LP831) embryos. The surfaces of internalizing EPCs that were not covered by other cells in 3 dimensions are outlined, and the black arrowhead points to these exposed EPC cell surfaces at the time of cell division, indicating a gastrulation defect that is not seen in the control.

51 homology groups: 27 groups that had transcripts that became enriched in two of the four internalizing cell types, 21 groups in three of the four internalizing cell types, and three groups enriched in all four internalizing cell types.

Of the three homology groups whose transcripts became enriched in all four internalizing cell types, two included genes encoding F-box domains (S1 and S2 Files). *C. elegans* has an unusually large number of F-box-encoding genes [45]. None of the genes in the two groups we identified has a known function or known orthologs outside of the Caenorhabditid nematodes. The third group consisted of three genes encoding proteins that have LIM domains; *lim-9* (an ortholog of LIMPET in *Drosophila* and FHL2 in vertebrates), *pxl-1* (an ortholog of paxillin in *Drosophila* and vertebrates), and the *zyx-1*/zyxin gene that we had identified in a separate set of experiments above.

Within this homology group, *zyx-1* transcripts were enriched in the E lineage as described above; *lim-9* transcripts were enriched in Cxp descendants (i.e., the cells of the C lineage that internalize) and not their non-internalizing Cxa sister cells; and *pxl-1* transcripts were enriched in MS descendants and D descendants (Fig 4C). These genes encode proteins whose homologs are involved in actin filament organization in muscle cells [46–49], focal adhesions and mechanotransduction [50,51], stretch-induced gene expression [52], planar cell polarity, and asymmetric cell division [53]. LIM domain-containing proteins have been shown to interact physically with components of the actin cytoskeleton such as vinculin and α-actinin [46]. We consider members of this homology group as interesting candidates for regulators of cell behaviors during gastrulation based on the transcript enrichments that we found, their broad conservation across animals, as well as their known involvement in cytoskeletal organization.

Therefore, we attempted to test whether these LIM domain-encoding genes are required during gastrulation in their respective cell lineages. We targeted each candidate individually by dsRNA injection, filmed embryos, and assayed for gastrulation defects among the internalizing cell lineages in which each candidate was found to be upregulated. Besides *zyx-1* RNAi (see Table 1), none of the candidates yielded gastrulation defects in their respective cell lineages (*pxl-1* RNAi, 0% MS or D lineage internalization defects, n = 24; *lim-9* RNAi, 0% C lineage internalization defects, n = 46). *C. elegans* also has another LIM domain-encoding gene, *unc-97*, that is expressed at high levels in internalizing C lineage cells and moderately high levels in other internalizing lineages (see Differential Expression Gene Explorer link above). Because complex genetic redundancy among a multi-gene family and/or failure to sufficiently knock down transcript levels by RNAi might have prevented us from observing phenotypes after *pxl-1* RNAi and *lim-9* RNAi, and because we had found that targeting *zyx-1/zyxin* did result in gastrulation defects, we decided to set aside work on the larger set of LIM domain proteins to focus on characterizing ZYX-1/zyxin's role in EPCs during gastrulation. Overall, our differential gene expression results led us to conclude that few genes can be found with expression patterns matching that expected for transcriptionally-regulated genes that might orchestrate apical constriction in diverse cell types, and it highlighted a possible role for LIM-domain-encoding genes including *zyx-1*.

We confirmed *zyx-1*/zyxin's role in gastrulation by generating a CRISPR knockout of *zyx-1* (*zyx-1Δ*, LP831), removing the protein coding region and replacing it with a cassette encoding a codon-optimized GFP expressed under the control of the *myo-2* promoter, driving expression in the pharynx to allow for easy visual identification of the allele. Consistent with the phenotype seen by RNAi, we also found gastrulation defects in the knockout strain. This phenotype was more penetrant than we had observed in the *zyx-1* dsRNA injection (41.3% vs. 25%, 19/46 Gad embryos, Fig 4D). To determine if AFD-1 and ZYX-1 might function redundantly, we performed *afd-1* RNAi in the *zyx-1Δ* strain and found that *afd-1* RNAi did not increase penetrance (9/36 Gad, 25%).

Before examining whether ZYX-1 is involved in linking junctions to contracting apical actomyosin networks, we attempted to characterize its localization. We anticipated that the low level of *zyx-1*/zyxin transcripts that we detected in EPCs might make it difficult to visualize protein localization; indeed ZYX-1 might function specifically at a low level, during a time when levels are only beginning to rise transiently after initial gene expression. Previous authors have reported that *zyx-1*/zyxin produces 2 protein isoforms: a longer, 603 amino acid isoform called ZYX-1a, and a shorter, 200 amino acid isoform called ZYX-1b [46]. ZYX-1a contains 3 polyproline-rich repeats, a predicted nuclear export signal, and 3 tandem LIM domains (S4A Fig). We created a strain with mNeonGreen (mNG) inserted at the endogenous N-terminus to tag ZYX-1a, but consistent with its low predicted expression at this stage of development, we were unable to detect mNG signal in the EPCs, and we were unable to detect ZYX-1 by immunostaining methods that employed signal amplification (S4B and S4C Fig). In young adults we could see mNG::ZYX-1a readily in differentiated body wall muscle, neurons, gonads, and spermatheca, in line with where its expression was previously described [46,54] (S4D Fig). To assess where ZYX-1 could associate when accumulating in gastrulating cells, we examined where overexpressed ZYX-1 would localize using single-copy transgenes driven by the *sdz-1* promoter (P*sdz-1*), which is predicted to drive ~20-fold overexpression compared to *zyx-1* expression levels in EMS, E, and MS cell lineages (S5A Fig). We created two mNG-tagged constructs: one expressing full length ZYX-1a, and another expressing only the LIM domain-containing region (LCR) of ZYX-1 to examine where the LCR alone could direct localization. For both constructs, cytoplasmic mNG signal could be detected in E and MS cells as predicted, and small foci could be detected at the apical surfaces of internalizing EPCs (S5B Fig). Additionally, the predicted nuclear export signal of ZYX-1a appeared to be functional in full-length mNG::ZYX-1a, because mNG::ZYX-1a was excluded from the nucleus while mNG::LCR$^{ZYX-1}$ was not (S5C Fig). We conclude that ZYX-1a is likely expressed normally at too low a level as EPCs internalize to detect by current methods, and that it and its LCR can be recruited to apical foci in EPCs when overexpressed. One hypothesis consistent with our expression data and our phenotype data is that zyxin may be a limiting component required for triggering apical constriction that is expressed only briefly and at a low level at the onset of cell internalization. We attempted to test this hypothesis using the P*sdz-1* overexpression construct to drive expression in the EMS cell (which produces both the E and MS lineages) to see if expressing zyxin early and at higher than normal levels might result in early cell internalization, but we did not see cell internalization occurring earlier in this strain (S6 Fig). In line with *zyx-1*'s expression enriched in only EPCs, P*sdz-1*-driven overexpression of full length mNG::ZYX-1a was able to rescue most of the defects seen in the *zyx-1Δ* background (2/27 Gad, *p* = 0.0003). Overexpression of mNG::LCR$^{ZYX-1}$ was not able to rescue similarly (7/22 Gad, *p* = 0.366). We conclude that ZYX-1 contributes to gastrulation, and that domains beyond the LIM-domain-containing region are important for this function. Next, we investigated whether ZYX-1/zyxin and AFD-1/afadin contribute specifically to coupling cell junctions to contracting actomyosin networks during gastrulation.

## Development of a semi-automated image analysis workflow to quantify the degree to which myosin and membrane movements are coupled *in vivo*

To determine whether ZYX-1/zyxin and AFD-1/afadin are required for coupling actomyosin to cell junctions in EPCs, we constructed a dual-labeled strain to visualize myosin particles as well as plasma membranes, to serve as a proxy for cell-cell junctions. We used an existing strain with endogenously-tagged NMY-2/myosin bearing an N-terminal fusion to an mNeon-Green (mNG) fluorophore [55] and inserted a single-copy transgene containing the bright red fluorophore mScarlet-I (mSc) [56] fused to the pleckstrin homology domain from phospholipase C-δ1 (PH domain), which localizes to plasma membranes. We then collected dual-color 4D confocal videos of membrane and myosin dynamics throughout the process of apical constriction (Fig 5A).

Previously reported methods for quantifying actomyosin and junction movements used a manual analysis of a relatively small number of datapoints, with a significant time investment [13]. To increase throughput and to ensure unbiased analysis, we developed a semi-automated pipeline to assess membrane and myosin movements. Because this is a new and potentially widely-applicable pipeline that required new code (freely available; see Materials and methods), we describe briefly below each of three steps that we used: (1) cell segmentation, (2) myosin flow computation and (3) analysis of the correlation between membrane and myosin movement. First, to calculate membrane movement we performed coarse segmentation of the Ea and Ep cells using Labkit [57] to train a random forest classifier to

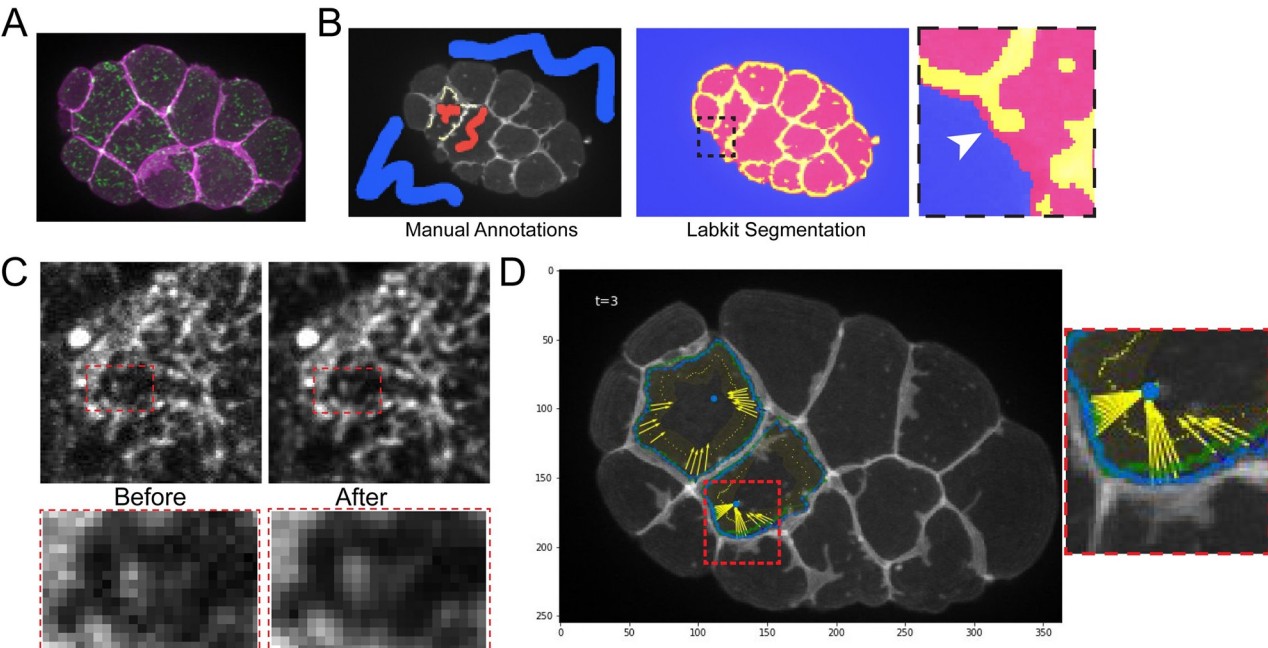

**Fig 5. Semi-automated workflow for analyzing actomyosin-membrane coupling. (A)** Z-projection of spinning-disk confocal images of ventrally-mounted embryos with labeled membrane (mScarlet-I::PH) and myosin (mNG::NMY-2). **(B)** Cell segmentation using Labkit in Fiji trained with user defined annotations for background (blue), cell interior (red), and membrane borders (yellow) for 3 time points (beginning, middle, and end) (left). Labkit outputs (right) were manually corrected as needed (inset, white arrowhead, see methods). **(C)** Noise2Void was used to clean up the mNG::NMY-2 images to assist with the automatic flow detection. **(D)** Image showing membrane segmentation (green and blue outlines), centroid (blue dot) and centripetal vectors calculated from myosin flows (yellow arrows). The shaded yellow region indicates the 2 μm border region mentioned in the main text.

recognize (i) cell interior, ie. pixels in the apical cortex's interior, (ii) cell membrane (border), and (iii) background, ie. everything else. For each time-lapse, 1–2 minutes were required to draw a few manual annotations for each pixel class and train the random forest classifier (Fig 5B, left). Automatic segmentations were then manually reviewed and adjusted to ensure that the borders of Ea and Ep cells were properly segmented (Fig 5B, right). These adjustments were especially needed where Ea or Ep had no neighbors visible in the imaged plane, to prevent incompletely segmented objects from merging into one another (Fig 5B inset, arrowhead). Total segmentation and curation time was on average about 10–15 minutes per film.

Second, we reduced image noise and computed myosin flows. We denoised the movies with a self-supervised denoising method, Noise2Void [58]. We manually checked several randomly selected time points to ensure that the denoising performed well (Fig 5C). We then generated a maximum intensity Z-projection and used these for myosin flow quantification. We computed myosin flows using the Farnebäck Optical Flow [59] implementation in OpenCV [60], only considering the optical flow for myosin particles that lie along vectors drawn between the membrane and either the cell centroid or centers of myosin flow (Fig 5D). We hereafter refer to such vectors as centripetal vectors. Additionally, we considered only the purely centripetal components of these vectors, since only this component contributes to movement of the membrane in this direction (Fig 5D, yellow arrows). To limit our analysis to only actomyosin that might contribute to coupling at cell junctions, we defined a two-micron window at the outer border of the centripetal vectors near the cell border for each time point and only considered myosin flows that fell in this region and were moving centripetally (Fig 5D, yellow shaded region).

Third, we analyzed the degree to which myosin and membrane movements were coupled. For each pair of time points and for each centripetal vector, we calculated the net membrane movement vector and the net myosin movement vector (average of all centripetal myosin vectors). This calculated difference in the movement of membrane and myosin is defined as the slippage rate, as described previously [13].

We found that this analysis recapitulated the finding by Roh-Johnson et al. 2012 that there is slippage between myosin and membrane movements in wild-type embryos, as well as a significant decrease in such slippage over time, but with many more data points than possible previously (Fig 6A and 6B and Table 2). Overall we found lower myosin and slippage rates than previously; this is not unexpected, because our new workflow enables analysis of optical flow that considers smaller myosin particles than before, whose movements would be expected to be more strongly randomized by brownian motion.

## Zyxin affects coupling of cell junctions with contracting actomyosin networks

Having developed a semi-automated method for analyzing myosin and membrane movement that confirmed previous findings, we sought to determine whether AFD-1/afadin and ZYX-1/zyxin were required for membrane to move centripetally along with actomyosin contractions. We used our strain labeled with mNG::NMY-2 and mSc::PH and performed RNAi targeting *afd-1*. We performed experiments involving *zyx-1* using our CRISPR-generated *zyx-1 Δ*, which we crossed into the dual-labeled strain.

We first examined whether *afd-1* or *zyx-1* had unanticipated effects on myosin dynamics. The observed rates of myosin movement appeared to be largely unaffected by the loss of either *afd-1* or *zyx-1*: The mean myosin velocities early (E) and late (L) measured by our image analysis method across all conditions were similar to one another (Fig 6A and Table 2). These

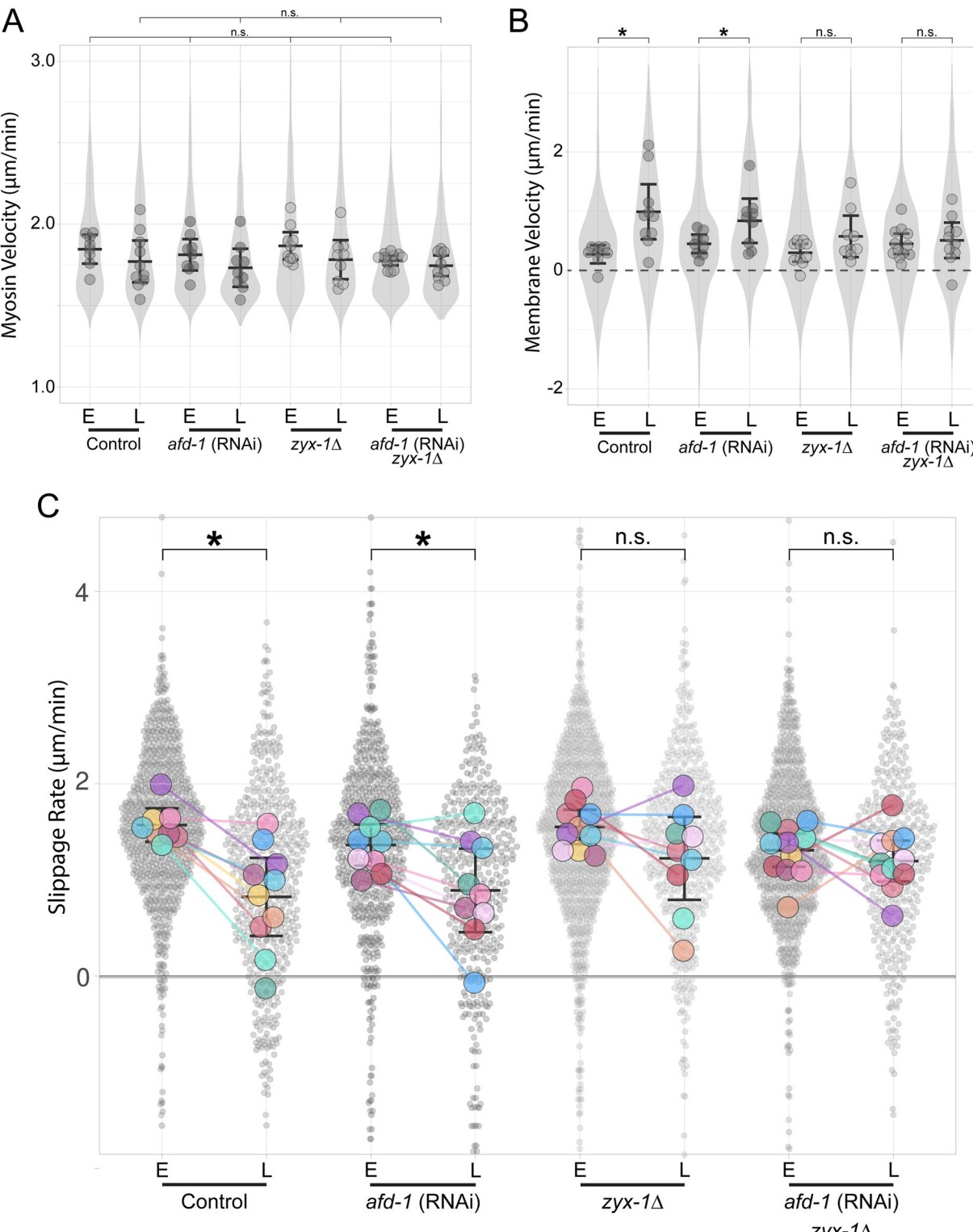

**Fig 6.** *zyx-1*/**zyxin is required for proper myosin/membrane movements during apical constriction.** Mean data for each replicate shown as a larger dot. Data were binned into early (E) or late (L) stages with paired-data in (C) indicated by a colored line to highlight the difference between E and L for paired replicates. **(A)** Measurements of myosin velocities along centripetal vectors at both E and L stages are similar across all conditions. No pairwise comparison was significantly different between E and L stages for wild-type (control) embryos or for any of the experimental treatments. **(B)** Measurements of junctional membrane movement rates along centripetal vectors at both E and L stages show

significant differences for control and *afd-1* RNAi ($p < 0.005$) but not in *zyx-1 Δ* ($p = 0.105$) and double ($p = 0.699$). **(C)** Superplot [88,93] of calculated slippage rates between myosin and membrane movement with individual centripetal vector measurements displayed as small semi-transparent points. Mean data for each replicate shown as a larger dot. Slippage rates for all E stage samples except the double were statistically indistinguishable ($p = 0.019$). The only samples with a significant difference in slippage between E and L were the control ($p < 0.002$) and *afd-1* RNAi ($p < 0.05$). Complete data quantification shown in Table 2. For visualization, 18 data points between 4.5 and 6.0 μm/min and 8 data points between -2 and -3 μm/min were not shown although they were included in the analysis. Means for each condition are indicated by black bars with error bars representing 95% confidence intervals.

results confirmed to us that *afd-1* and *zyx-1* are unlikely to affect apical constriction of EPCs through large, unanticipated effects on myosin velocities.

Next, we examined whether the normal inward movement of junctions that arises gradually over time depended on *afd-1* or *zyx-1*. Comparison of mean membrane velocities between early and late stages revealed an acceleration of inward membrane movement in both the control and *afd-1* RNAi embryos that was largely lost in *zyx-1Δ* or *zyx-1Δ+afd-1* RNAi embryos (Fig 6B and Table 2). As expected given these results and the negligible effect of these genes on centripetal myosin velocities, *afd-1* RNAi did not prevent the normally significant loss of slippage over time, but *zyx-1Δ* or *zyx-1Δ+afd-1* RNAi did (Fig 6C and Table 2). *zyx-1 Δ + afd-1* RNAi embryos displayed coupling defects in mean slippage rates between early and late stages that were not significantly different ($p = 0.891$) than *zyx-1 Δ* on its own (Fig 6C and Table 2). Taken together, we conclude from the results that ZYX-1 contributes directly or indirectly to the degree to which cell-cell junctions move with the contracting actomyosin cytoskeleton.

## Discussion

In cells undergoing apical constriction, cortical actomyosin contractions can begin before the apical sides of cells constrict [13]. This finding suggested that apical constriction may be triggered by regulating connections between already-contracting apical actomyosin networks and

**Table 2. Summary of data and statistical tests for myosin velocity, membrane movement rate, and slippage differences between early (E) and late (L) stages.**

| Condition | Measurement | Mean (μm/min) | | Mean Difference | 95% C.I. | | Early vs. Early Control | Late vs. Late Control | Early vs. Late | N (E / L) |
|---|---|---|---|---|---|---|---|---|---|---|
| | | Early | Late | | Early | Late | *p*-value | *p-value* | *p*-value | |
| *Control* | Myosin velocity | 1.85 | 1.77 | 0.08 | 0.09 | 0.13 | - | - | 0.2538 | 8 / 10 |
| *afd-1 (RNAi)* | Myosin velocity | 1.81 | 1.73 | 0.08 | 0.10 | 0.12 | 0.5248 | 0.5998 | 0.2130 | 9 / 9 |
| *zyx-1 ko* | Myosin velocity | 1.87 | 1.78 | 0.09 | 0.08 | 0.12 | 0.6907 | 0.8758 | 0.1763 | 10 / 9 |
| *afd-1 (RNAi) + zyx-1 ko* | Myosin velocity | 1.78 | 1.75 | 0.03 | 0.03 | 0.06 | 0.0983 | 0.6738 | 0.2498 | 12 / 11 |
| *Control* | Membrane movement | 0.27 | 0.99 | 0.72 | 0.16 | 0.46 | - | - | 0.0048 | 8 / 10 |
| *afd-1 (RNAi)* | Membrane movement | 0.45 | 0.84 | 0.39 | 0.16 | 0.37 | 0.0637 | 0.5492 | 0.0038 | 9 / 9 |
| *zyx-1 ko* | Membrane movement | 0.30 | 0.57 | 0.27 | 0.15 | 0.35 | 0.7712 | 0.1055 | 0.1048 | 10 / 9 |
| *afd-1 (RNAi) + zyx-1 ko* | Membrane movement | 0.45 | 0.51 | 0.06 | 0.17 | 0.30 | 0.0841 | 0.0547 | 0.6991 | 12 / 11 |
| *Control* | Slippage | 1.57 | 0.82 | 0.75 | 0.18 | 0.41 | - | - | 0.0016 | 8 / 10 |
| *afd-1 (RNAi)* | Slippage | 1.36 | 0.89 | 0.47 | 0.22 | 0.44 | 0.0822 | 0.7865 | 0.0351 | 9 / 9 |
| *zyx-1 ko* | Slippage | 1.55 | 1.22 | 0.33 | 0.18 | 0.44 | 0.8479 | 0.1204 | 0.1184 | 10 / 9 |
| *afd-1 (RNAi) + zyx-1 ko* | Slippage | 1.31 | 1.20 | 0.11 | 0.17 | 0.21 | 0.0193 | 0.0739 | 0.3510 | 12 / 11 |

apical junctions. Here, we sought to identify molecules that could contribute directly or indirectly to such connections. We used proteomic and transcriptomic approaches to identify new genes that contribute to normal gastrulation in *C. elegans*. Among the genes we identified were two genes, *afd-1*/afadin and *zyx-1*/zyxin, that encode proteins of families already known to indirectly connect actin networks and membrane proteins in other contexts, but which had not been implicated in the kind of temporally-regulated linkage during normal development that we sought here. We found that *zyx-1*/zyxin was required for the normal reduction of slippage over time between junctions and contracting actomyosin networks. Our work also produced two new resources available to other researchers: transcriptomes of multiple gastrulating lineages from *C. elegans* available in interactive online form to facilitate querying the dataset, and a semi-automated method for analyzing myosin and membrane movements. Together, our results identify new genes that contribute to *C. elegans* gastrulation including one encoding a key protein that may contribute directly or indirectly to a previously-hypothesized molecular clutch for apical constriction [13].

Our results suggested that AFD-1/afadin makes either minor, redundant, or no contributions to the dynamic regulation of coupling that we sought. Afadin has known functions in apical constriction in *Drosophila*, including connecting membrane proteins to F-actin [27], supporting junctional integrity [61], and maintaining apical/basal polarity [32]. Afadin was first identified as an actin-binding protein localized to CCC-based adherens junctions in mice [22]. Prior to this study in *C. elegans*, *afd-1* had been shown to genetically interact with *sax-7*/L1CAM during gastrulation [30]. Although the precise role of *sax-7*/L1CAM is not well understood during gastrulation in *C. elegans*, *sax-7* is partially redundant with *hmr-1*/cadherin [34], and afadin has been found in other systems to interact with both nectin, which is another immunoglobulin cell-adhesion molecule (IgCAM) not directly related to *sax-7*, [62] and CCC adhesion complexes [36,37]. Therefore, it is possible that afadin has conserved interactions with both CCC-based and IgCAM-based adhesion, and loss of afadin might affect both to some degree. Our data suggest that AFD-1 localization at junctions in *C. elegans* depends on HMR-1/cadherin but not HMP-1/α-catenin or SAX-7/L1CAM, which is surprising given previous literature in other systems that suggests an interaction between afadin and α-catenin [36,37]. It is possible that multivalent interactions between multiple junctional proteins [63,64] contribute to recruiting AFD-1 to junctions.

Because gastrulation in *C. elegans* relies on embryonic transcription [20], we hypothesized that some regulators of *C. elegans* gastrulation might be identified by analyzing cell-specific changes in transcriptomes. Our single-cell transcriptome analysis identified a group of LIM domain-containing proteins as specifically upregulated in internalizing cells. LIM domains are composed of tandem zinc fingers and share a few conserved residues required for the coordination of zinc ions [47,65]. Proteins containing LIM domains play evolutionarily conserved structural roles supporting actin networks under stress [66,67]. Some LIM proteins like LMO7 might directly regulate apical myosin activity [68]. Smallish is an LMO7 homolog in flies that localizes to the zonula adherens, binds to PAR polarity complex proteins as well as canoe/afadin, and regulates morphogenesis and actomyosin contractility [69]. Because our LIM domain-containing candidate proteins were specifically enriched just before the onset of cell internalization, we hypothesized that apically constricting cells may rely on different LIM domain-containing proteins to successfully internalize. Although we found that RNAi of only one LIM domain-containing candidate had an effect on cell internalization, we cannot rule out the possibility that genetic redundancy or incomplete RNAi penetrance contributed to the lack of an observable phenotype after targeting two other LIM domain genes that we identified. Genetic redundancy is common among morphogenesis mechanisms in various systems [70–72].

The LIM domain-containing protein that had a measurable effect on cell internalization was *zyx-1*/zyxin, which is expressed in EPCs. Zyxin is a tension-sensitive protein that can bind actin filaments under stress and contribute to repair of actin stress fibers [25,51]. In humans, the zyxin-family of LIM proteins contains several members, including Ajuba, WTIP, TRIP-6, and others. There is substantial precedent for zyxin-like proteins contributing to apical constriction in other systems, by incompletely defined mechanisms. WTIP is important for apical constriction in *Xenopus* [73], where it localizes to junctions. In human keratocytes, Ajuba localizes to junctions via an interaction with α-catenin [74] and is required for Rac activation and E-cadherin-dependent adhesion [75]. In *Drosophila melanogaster*, Ajuba has also been shown to localize to junctions via interaction with α-catenin that is dependent on myosin-induced tension [76,77]. ZYX/zyxin is the sole zyxin family member in *C. elegans*; it can contribute to the repair of junctional actin networks, and it is required for proper muscle function [46,78]. Endogenous tags of *zyx-1* were too dim to be visualized, and immunostaining experiments failed to detect signal as well, likely because expression in the EPCs is too low. Therefore, we do not know if ZYX-1 normally localizes to junctions and/or to actin networks in EPCs, leaving the mechanism by which it affects gastrulation and apical constriction an open question. We speculate that ZYX-1 might normally localize to junctions and indirectly strengthen mechanical connections between junctions and contracting actomyosin networks. Alternatively, ZYX-1 might function throughout the apical actin cortex and contribute to coupling by enabling rapid repair of stressed actin filaments.

Our study reveals new genes that contribute to *C. elegans* gastrulation. Given the nature and penetrance of the phenotypes we observed after loss of *afd-1*/afadin and *zyx-1*/zyxin, we suspect that multiple partially redundant mechanisms ensure successful cell internalization. Our single-cell transcriptome data should provide a wealth of data for comparing internalizing cell lineages to their non-internalizing neighbors and yield future insights into mechanisms. Numerous links between CCC, actin, and zyxin/LIM domain-containing proteins have been demonstrated in a variety of systems. Our study suggests that these same proteins might be working together in *C. elegans* specifically to link the forces generated by actomyosin contraction to cell-cell junctions during apical constriction.

## Materials and methods

### *C. elegans* maintenance and strains

Nematodes were cultured and handled as previously described [79]. Strains used in this study are indicated in S1 Table.

### Co-immunoprecipitation and mass spectrometry

*C. elegans* expressing either soluble GFP alone as a control or HMP-1::GFP were used for affinity purification of GFP from isolated embryos. Liquid culture and affinity purification of protein complexes from *C. elegans* extracts were performed following previously described methods [80]. Briefly, we isolated embryos from unsynchronized liquid cultures by bleaching and incubated them overnight in M9 media to obtain synchronized L1 larvae, which were then used to start new liquid cultures. Synchronized cultures were allowed to grow until the majority of *C. elegans* were observed to reach adulthood and contain ~5 embryos, to enrich for embryos with <50 cells. At this point, embryos were again harvested by bleaching, washed in M9, transferred to lysis buffer adapted from [81] (20 mM Tris-HCl, pH 7.9; 150 mM NaCl; 10% Glycerol; 1.0 mM; 0.5 mM DTT; 0.05% Triton-X 100; 1 Complete EDTA-free Protease Inhibitor cocktail tablet per 12 mL lysis buffer (Roche Applied Science,

#1873580)) and drop frozen in liquid nitrogen. Embryonic extracts were prepared using pulverization and sonication followed by a single-step immunoprecipitation using anti-GFP coupled agarose beads (MBL international, D153-8). Purified protein extracts were submitted for LC-MS/MS analysis at the UNC Michael Hooker Proteomics Center on a Thermo QExactive HF machine (ThermoFisher).

## Candidate gene selection

An initial list of 545 candidate proteins from the affinity purification of HMP-1::GFP from embryos was filtered first by removing 315 proteins that shared peptide hits in the soluble GFP control. The remaining 230 proteins all had at least 1 peptide hit. Both HMP-2/β-catenin and HMR-1/cadherin were identified in this pool of candidates with the highest peptide counts, suggesting that our method worked. The remaining candidates were further filtered by ribosomal proteins, elongation factors, and other common housekeeping proteins which reduced the list to 126 proteins. Finally, because synchronization of *C.elegans* is not perfect we only considered candidates with evidence of gene expression in EPCs [19], which reduced our list to 11 proteins expressed in the early embryo with a potential physical interaction with the CCC. Candidates from transcriptome data were initially selected if they showed evidence of transcript enrichment in the EPCs at stages just prior to the onset of apical constriction (at either the 8-cell or 24-cell stage of development). 21 candidate genes were selected from among those that have at least an 8-fold (i.e. log2 of 3) enrichment in the EPCs at either of those two stages. Enrichment for each gene was calculated after removing the EPC(s) contribution to the whole embryo total.

## RNA interference

Primers were designed to amplify ~1000 bp of each target genes' protein coding sequence (S2 Table). Each primer also included 15 bases of the T7 promoter sequence at the 5' end to be used in a 2-step PCR from wild-type genomic DNA sequence. The PCR product was purified using a Zymo DNA Clean and Concentrator kit (Zymo Research) and used as a template for another round of PCR using primers containing the full-length T7 promoter sequence. After a second purification using a Zymo DNA Clean and Concentrator kit the PCR product was used as a template in a T7 RiboMAX express RNAi System (Promega) following the manufacturer's protocol. Purified dsRNA was injected at a concentration of 500 ng/μL into L4 or young adult hermaphrodites using a Narishige injection apparatus, a Parker Instruments Picospritzer II, and a Nikon Eclipse TE300 microscope with DIC optics. Excess dsRNA was stored at −80˚C. Injected worms were allowed to recover on a seeded NGM plate for 24–36 hours at 20˚C before harvesting embryos for imaging.

## Assay for gastrulation defects

Gastrulation assays were performed as previously described [23]. Briefly, *C. elegans* embryos were dissected in Egg Buffer from dsRNA injected gravid adults and mounted on poly-L-lysine coated coverslips, supported by a 2.5% agarose pad. Four-dimensional differential interference contrast (DIC) microscopy was performed using a Diagnostic Instruments SPOT2 camera mounted on a Nikon Eclipse 800 microscope. Images were acquired at 1 μm optical sections every 1 minute during embryogenesis and analyzed using ImageJ [82]. Embryos were considered gastrulation defective (Gad) if either Ea or Ep divided before it being fully internalized. Imaging was performed at 20˚C—23˚C for all strains.

## CRISPR editing

Strains were created using previously reported methods [55,83]. Proteins were tagged on either their N- or C-termini after considering the presence of multiple isoforms or knowledge of existing tags in other organisms. For afadin, both N- and C- terminally tagged transgenes have been used previously in other organisms without any reported defects and N-terminally tagged transgenes can rescue lethal phenotypes in flies [33,84,85]. Repair templates were first constructed by inserting 500–1000 bp of homologous sequence amplified from genomic worm DNA into a vector on either side of a fluorescent protein and a selection cassette using Gibson Assembly or SapTrap methods [86,87]. For the *zyx-1* gene deletion, the homologous sequence was inserted into a vector containing *myo-2* promoter driven GFP as a visible marker of the deletion. Cas9 guide sequences were selected using the CRISPR Design tool (crispr.mit.edu, no longer available) and cloned into Cas9-sgRNA expression vector DD162 [83] and then co-injected into adult germlines along with the repair template vector and array markers. Selection of edited worms was conducted using previously described methods [55].

## Fluorescence imaging

Laterally mounted embryos were imaged on 2.5% agarose pads, ventrally mounted embryos were imaged using clay feet as spacers between the slide and coverslip. Embryos were imaged using a spinning disk confocal microscope with a Nikon TiE stand and a 60X 1.4NA Plan Apo immersion oil objective (Nikon), CSUXI spinning disk head (Yokogawa), and an ImagEM EMCCD (Hamamatsu). For analysis of coupling, images were collected in sets where the membrane was imaged on the 1st and 7th frames, and myosin was imaged in every frame. Optical sections of 0.5 μm were collected to a depth of 2 μm from the surface of the embryo. In doing this, a membrane volume was collected every ~34.3 seconds while myosin volumes were collected every ~5.7 seconds. Z-projections were analyzed using ImageJ and our automated analysis pipeline (see Methods below).

## Quantification of mKate2:AFD-1 levels at junctions

The analysis of apical enrichment of mKate2::AFD-1 in Ea/p was performed as described previously for members of the CCC [15]. The analysis of relative mKate2::AFD-1 levels along junctions was performed on embryos that were isolated from adult worms 24 hours after being injected with dsRNA. Embryos from worms injected with either *hmr-1* or *hmp-1* dsRNA were mounted side-by-side on agar pads with control and oriented such that embryos could be differentiated on the microscope. mKate2::AFD-1 junctional intensity was collected by taking stacks of images 12 microns deep into the embryo. Z-projections of embryos were analyzed using ImageJ to calculate the average intensity along a 50 pixel long, 5 pixel wide line drawn along identical junctions in both the control and experimental conditions. Average fluorescence intensities (fluorescence intensity) were then adjusted by subtracting off-embryo background levels by drawing the same 50 pixel long, 5 pixel wide line in a space adjacent to the embryos for each prepared slide. Adjusted average pixel intensities were normalized to the highest value in each experimental group and plotted to compare Control vs. Experimental junctional intensities. Ratios from each group were calculated by first taking the ratio of the average pixel intensities from each embryo pair, and then plotting them along with the group averages. Statistical analysis of ratios was performed using a Welch's t-test in the Superplots webtool (https://huygens.science.uva.nl/SuperPlotsOfData/, [88]). mKate2::AFD-1 embryos showed a low-penetrance gastrulation defect (1/24 embryos), raising the possibility that our N-terminally tagged-protein had minor effects on its function; this was significantly weaker penetrance than the defects seen by RNAi ($p = 0.018$).

## Worm dissections

Worms were grown and dissected, and RNA was prepared and sequenced, as in Tintori 2016 [19]. Briefly, embryos were selected 10–20 minutes before the desired stage, chemically disrupted using a sodium hypochlorite solution and a chitinase/chymotrypsin solution, then mechanically disrupted by agitation via mouth pipette. The only exceptions were that in this study embryos were dissected manually by aspiration on a Yokogawa spinning disk confocal microscope under brightfield illumination, and cell types were identified by fluorescent markers, as illustrated (S1 Fig).

## Single embryo transcriptomes and analysis

We chose to isolate cells expressing lineage markers from individual embryos of known stages so that we could be confident of staging and cell identity, and to ensure that we had isolated fully intact cells (because unlysed cell fragments have been seen on occasion after manual shearing, suggesting that shearing can sometimes result in bisection of cells) and without reversing incomplete cytokinesis by isolating cells before cytokinesis was complete [89]. The transcriptomes already available from our previous study had been generated from single cells [19]. To keep any amplification artifacts consistent across datasets, we chose to perform RNA-seq on later time points using material from single embryos rather than bulk-isolated cells. For the same reason, we used the same kit and protocol that had been used for earlier samples [19]. Briefly, cDNA was generated using the SMARTer Ultra Low RNA Input for Illumina Sequencing Kit, and sequencing libraries were prepared using the Nextera XT kit. Sets of samples from a single embryo were rejected if one or more libraries had an over-representation of ERCC spike in reads (if ERCC spike in transcripts were more than 1/10th as abundant as worm transcripts) [90].

With the addition of these new datasets, we have now sequenced transcriptomes for all cell types of the 1-, 2-, 4-, and 8-cell stage, 9 cell types of the 24-cell stage (ABalx, ABarx, ABplx, ABprx, MSx, Ex, Cx, D, and P4), and 5 groups of cells from the 100 cell stage embryo (ABp descendents, "ABp+"; MS descendents, "MS+"; D descendents, "D+"; and internalizing and non-internalizing C descendents, "C-in" and "C-out"), as well as the partially overlapping groups of cells that make up the rest of the embryo from each of those 100 cell stage dissections (e.g. all non-D cells, "D-", to match each set of D cells collected, "D+"). An approximation of the remainder of the 100 cell stage embryo, referred to here as ABa-E-P4, was generated in silico based on weighted averages of other samples from that stage using the following approach. First, a whole embryo average was calculated based on the weighted averages of all dissections from the gastrulating stage (for example, 0.25 x RPKMs for ABp+ plus 0.75x RPKMs for "ABp-", with weightings based loosely on mass of each cell cluster). We then subtracted the weighted values of each of the targeted cell types (subtracting 0.25x"ABp+", 0.0625x"C-in", 0.0625x"C-out", 0.0625x"D+", and 0.125x"D+") from the whole embryo estimate, leaving an estimate of just the ABa-E-P4 RPKM values. A previous study generated transcriptomes for many of these cell types by isolating founder cells starting at the 2- cell stage, and allowing cells to divide in culture before collecting them for RNA-seq [91]. The method used in that study kept cells naïve to critical cell-signaling events. We were interested in preserving those fate-determining signaling events, so we collected cells based on fluorescence within 10 minutes of being disrupted from their native environment in the embryo.

Transcripts were considered "detected" if their RPKM value was above a threshold of 25. All fold change calculations were done on adjusted RPKM values—raw RPKM values with 25 added to them—to avoid enriching for small differences between samples with low RPKM values.

Among 57 replicates of 9 samples from the 100 cell stage, we detected transcripts from 7,998 genes (above a threshold of 25 reads per kilobase of transcript per million mapped reads, or RPKM). This value roughly matched our expectations based on transcriptomes generated from earlier cell types—in our previous study we detected 8,575 genes amongst 1- to 24-cell stage embryos (note that owing to division asynchrony between lineages, there is no 16-cell stage in *C. elegans*, but the 16 samples from the 24-cell stage were referred to for convenience as cells of the 16-cell stage in our previous study). In the previous study we thoroughly validated our low-input RNA-seq data by (1) comparing them to previously known gene expression patterns and (2) comparing them to single molecule fluorescent *in situ* hybridization assays [19]. The sequencing technology used in this study was virtually identical, with the main difference being that the samples collected were from groups of smaller cells later in development, rather than single larger cells earlier in development. To validate the dataset in the present study, we compared our mRNA sequencing data to protein level data for the Cxa (C-out) and Cxp (C-in) samples, as previously reported [92]. We used the EPIC database (https://epic.gs.washington.edu/) from Murray et al. 2012 to identify proteins that are differentially expressed between Cxa cells and Cxp cells (e.g. elt-1, nhr-171, and vab-7, S2 Fig), and inspected our transcriptome data for matching trends. We chose these samples because they were the most technically difficult dissections, due to the cells' small size and low fluorescence levels, and hence were the samples we had the least confidence in.

Families of proteins were defined by creating groups of genes based on similarity, using a protein BLAST E-value cutoff of $e^{-15}$.

## Analysis of slippage

Slippage is defined as the difference in velocity between myosin particles and the adjacent membrane [13]. These rates are measured along the centripetal vectors, which we refer to as "Spyderlegs" in our code. To reduce noise from brownian motion or other non-myosin movement we used a filter to remove flow vectors moving slower than 1.5 μm/min which is slightly slower than reported myosin velocities in these cells [13]. This filter is adjustable in the pipeline so it can be tailored for use in other systems. The cell centroid (blue dots in Fig 4D) was initially seeded manually and then automatically determined for each subsequent frame. In cases where the myosin flow was coalescing on an off-center point, the automatically determined center was manually overridden. A slippage rate of 0 means that both membrane and myosin are moving in concert with each other, while a positive slippage rate signifies that the myosin velocity is higher than the adjacent membrane. The values obtained from this semi-automated image analysis pipeline were then binned into two categories, early and late, relative to the birth of the neighboring MSxx cells. Early stages were defined as being between 3 and 7 minutes post MSxx birth, while late stages were 13 minutes post MSxx birth and later. Slippage rates were then plotted using Superplots (https://huygens.science.uva.nl/SuperPlotsOfData/, [88]).

The semi-automated analysis pipeline described in the Results determines the slippage rate over time for each centripetal vector (Spyderleg) as well as individual cell (Ea and Ep) averages and can be completed in about 30–45 minutes per embryo. While there is still room for further automation, the presented analysis pipeline is an important step toward making the required analyses feasible on a large scale, while still offering users the possibility to override erroneous automated decisions.

## Statistical analysis

Comparative analyses of Gad phenotypes between different groups were performed using Chi-square goodness of fit test. All other analyses were performed using a Welch's t-test, or paired t-test unless otherwise noted.

## Supporting information

**S1 Fig. Dissection of multiple divergent, gastrulating cell types.** Four fluorescent marker strains that were used to dissect and collect each sample collected from the 100 cell stage.
(TIF)

**S2 Fig. Validation of C descendent transcriptomes.** Three genes with protein expression in C descendants were selected from the Waterston Lab's lineagomics database (epic.gs.washington.edu) and compared to our transcriptome data. Cell lineages show only the C descendants, and are color coded by relative fluorescence levels detected from a film of embryonic development taken of embryos with multi-copy arrays of promoter fusions of each gene. According to the Waterston lineages (left), transcripts of *elt-1* and *nhr-171* are expected to be enriched in Cxa descendants, and transcripts of *vab-7* are expected to be enriched in descendants of Cxp, which we also see in our transcriptomes (right).
(TIF)

**S3 Fig. 11 candidates with close to complete internalization-associated expression patterns.** Pictograms showing individual cell heat map expression patterns of the genes indicated with RPKM values listed. None of these 11 candidates had expression patterns that perfectly matched with cell internalization, i.e. enrichment was weak in one or more groups of internalizing cells, or some non-internalizing cells showed strong expression.
(TIF)

**S4 Fig. Attempts to visualize endogenous ZYX-1 protein localization. (A)** *C. elegans* ZYX-1a consists of a few proline rich regions near the N-terminus (magenta boxes), a predicted nuclear export sequence (NES, orange box), and 3 tandem LIM domains. **(B)** *zyx-1* is expressed at the 8 cell stage in EPCs (19). Pictogram key in S4 Fig. **(C)** Endogenous mNG-tagged ZYX-1 does not accumulate in EPCs (white arrowhead) appreciably above background, although the expression levels (B) are predicted to be low at this stage. This image was taken with settings that amplified even low level fluorescence and background in an attempt to enhance any apparent signal. We also failed to detect ZYX-1 by immunostaining ZYX-1::GFP embryos and using TSA amplification. **(D)** Endogenous mNG-tagged ZYX-1 shows clear expression in young adults, with localization patterns matching previous reports for *zyx-1* transgenes (46), suggesting that our tagged gene is properly expressed.
(TIF)

**S5 Fig. Attempts to visualize ZYX-1 protein localization via overexpression in E and MS lineage cells. (A)** An mNG-tagged transgene driven by the *sdz-1* promoter, which drives expression in MS and E cell lineages ~20-fold higher than predicted levels for endogenous *zyx-1* (left). **(B)** mNG-containing puncta can be seen at the apical surface for both full length zyxin and the LIM domain-containing region (LCR) of ZYX-1. Apical slices at the depths indicated show more puncta closer to the apical surface, with fewer punta appearing further away from the surface. **(C)** The predicted NES is functional. If viewed from a central section, the LCR$^{ZYX-1}$ construct, which lacks the NES, is not excluded from the nucleus (yellow arrowheads).
(TIF)

**S6 Fig. ZYX-1 is likely not a sole limiting component for triggering cell internalization.** Plot of the time it took between MSxx birth and EPC internalization, in minutes. P*sdz-1* driven overexpression of neither mNG::ZYX-1 nor mNG::LCR$^{ZYX-1}$ affected the timing of cell internalization as compared to control embryos (ZYX-1, $p = 0.34$, n = 32; LCR$^{ZYX-1}$, $p = 0.65$, n = 14).
(TIF)

**S1 Table. Strain List.**
(TIF)

**S2 Table. RNAi Primers.**
(TIF)

**S3 Table. Proteomic Data.**
(TIF)

**S1 File. List of genes enriched in the indicated sets of internalizing cells.**
(TXT)

**S2 File. List of gene homology groups enriched in indicated sets of internalizing cells.**
(TXT)

## Acknowledgments

We thank members of the Goldstein lab, Jeff Hardin, Minna Roh-Johnson, and Mark Peifer for improving the manuscript. Some strains were provided by the Caenorhabditis Genetics Center (CGC; cbs.umn.edu/cgc/home), which is funded by NIH Office of Research Infrastructure Programs (P40 OD010440), and the National BioResource Project (NBRP; http://www.shigen.nig.ac.jp/c.elegans).

## Author Contributions

**Conceptualization:** Mark M. Slabodnick, Sophia C. Tintori, Bob Goldstein.

**Data curation:** Mark M. Slabodnick, Sophia C. Tintori, Mangal Prakash.

**Formal analysis:** Mark M. Slabodnick, Sophia C. Tintori, Mangal Prakash.

**Funding acquisition:** Mark M. Slabodnick, Sophia C. Tintori, Bob Goldstein.

**Investigation:** Mark M. Slabodnick, Sophia C. Tintori, Pu Zhang, Christopher D. Higgins, Alicia H. Chen, Timothy D. Cupp, Terrence Wong, Emily Bowie.

**Methodology:** Mark M. Slabodnick, Sophia C. Tintori, Mangal Prakash.

**Project administration:** Mark M. Slabodnick, Bob Goldstein.

**Resources:** Mark M. Slabodnick, Sophia C. Tintori, Florian Jug, Bob Goldstein.

**Software:** Mangal Prakash, Florian Jug.

**Supervision:** Mark M. Slabodnick, Florian Jug, Bob Goldstein.

**Validation:** Mark M. Slabodnick, Sophia C. Tintori, Pu Zhang.

**Visualization:** Mark M. Slabodnick, Sophia C. Tintori, Mangal Prakash.

**Writing – original draft:** Mark M. Slabodnick, Sophia C. Tintori, Bob Goldstein.

**Writing – review & editing:** Mark M. Slabodnick, Sophia C. Tintori, Bob Goldstein.

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
