## [Decision Letter · Decision Letter 0]

24 Aug 2022

Dear Dr Slabodnick,

Thank you very much for submitting your Research Article entitled 'Afadin and zyxin contribute to coupling between cell junctions and contractile actomyosin networks during apical constriction' to PLOS Genetics.

The manuscript was fully evaluated at the editorial level and by independent peer reviewers. The reviewers appreciated the attention to an important problem, but raised some substantial concerns about the current manuscript. Based on the reviews, we will not be able to accept this version of the manuscript, but we would be willing to review a much-revised version. We cannot, of course, promise publication at that time.

Should you decide to revise the manuscript for further consideration here, your revisions should address the specific points made by each reviewer. Following the reviewers' comments, additional experimental work and/or analysis (in addition to rewriting) will be expected. If you choose to resubmit to PLOS Genetics we will also require a detailed list of your responses to the review comments and a description of the changes you have made in the manuscript.

If you decide to revise the manuscript for further consideration at PLOS Genetics, please aim to resubmit within the next 60 days, unless it will take extra time to address the concerns of the reviewers, in which case we would appreciate an expected resubmission date by email to plosgenetics@plos.org.

[LINK]

We are sorry that we cannot be more positive about your manuscript at this stage. Please do not hesitate to contact us if you have any concerns or questions.

Yours sincerely,

Andrew D. Chisholm

Academic Editor

PLOS Genetics

Gregory Barsh

Editor-in-Chief

PLOS Genetics

Reviewer's Responses to Questions

**Comments to the Authors:**

Reviewer #1: Slobodnick et al describe their screen for factors that contribute to gastrulation and identify AFD-1 and ZXY-1 as factors contribute to gastrulation, in an effort to identify factors that link myosin contraction to the cadherin junctions and membrane. They also show that AFD-1 is cadherin-dependently enriched in Ea/Ep border, where cadherin localizes, with apical enrichment. The authors also develop and use a semi-automated image analysis pipeline to show that knockdown of either AFD-1 or ZXY-1 leads to a loss in the difference in “slippage” from early to late time points in the internalization/gastrulation process. These results are interesting and do implicate AFD-1 and ZXY-1 in gastrulation and slippage. However, as written the manuscript is somewhat incremental in advancing our understanding of these factors during gastrulation, and the authors also should provide a more in depth analysis of some of their data to better compared the mutant phenotypes to the control, as detailed below. As the authors point out in the introduction, Afadin already has been implicated in linking membrane proteins to actin, and the expression data on zyxin is based on over-expression. Without a more extensive analysis, the manuscript as written does not in this reviewer’s opinion provide an sufficient advance in our understanding of gastrulation for publication in PLOS Genetics.

Major comments.

1. The authors should explain how they paint with green pseudo color in Fig2B. Without an explanation, the figure is uninterpretable; why is the mutant cell a darker green and how did they do the pseudocoloring to indicate this? Figure 2A clearly shows the gastrulation defect , and thus Fig2B is not really needed but if included should be described in more detail.

2. The authors use over-expression to assess ZYX-1 localization with an mNeonGreen fusion. Given the over-expression caveat, an antibody that can detect endogenous ZYX-1 would be more convincing if such antibodies exist. At a minimum, the authors should show multiple focal planes if they want to assert that the detected over-expressed fusion is apically enriched; the supplemental figure panel only shows an apical section and no other sections, and thus one cannot assess any apical enrichment.

3. The authors never analyze gastrulation defects in a afd-1, zyx-1 double mutant, based on Nomarski images like those shown in Figure 2. In Figure 6, the authors show no significant difference between this double mutant and the single mutants with respect to myosin velocity or slippage. This is somewhat surprising and even suggests that they may act in the same pathway, as opposed to parallel pathways or contributions. The authors should assess the GAD phenotype as scored by Nomarski over time in the double mutants vs single mutants to verify that there is no additive or synergistic interaction.

4. The authors base their conclusion (that AFD-1 and ZYX-1 contribute to the linkage between myosin and the apical membrane/cadherin complex) on the loss of any difference in slippage between early and late time points during gastrulation. While this is an interesting observation, it is not clear to this reviewer how this implicates these proteins in this process, and the analysis could be much more extensive with respect to the comparisons of both early and late time points in the mutants, to the early and late time points in the control embryos. The authors compare early vs early slippage in each mutant vs control in Table 2 but do not provide the same analysis for late slippage. Indeed, inspection of Figure 6B and Table 2 indicates that the slippage rate may actually decrease in the double mutant relative to the control and single mutants (if the authors would examine more embryos); such a result would not be predicted for proteins that promote linkage between the apical membrane/junction and myosin. If anything the slippage rate should increase, and it appears it may decrease, as striking the p value for early mutant vs early control is much smaller for the double mutant compared to each single mutant. Moreover, the number of embryos analyzed is small (only 6, 6, 7 and 9). While the data points per embryo are large, embryo to embryo variability is a common phenomenon at the cell biological level in early embryos, and the authors need to increase the number of embryos analyzed to more carefully and conclusively assess any impact of the double mutant on early and late slippage rates.

Minor comments.

1. In Figure 1, the authors show that cadherin and beta-catenin co-localize and stay co-localized, whereas myosin and alpha-catenin start out co-localized but then diverge, with myosin moving in ahead of alpha-catenin. However, in the text the authors say that Figure 1 shows that both alpha and beta-catenin co-localize with cadherin, and that both lag behind myosin during constriction. The figures only shows beta-catenin co-localizing with cadherin (not alpha-catenin) and only shows alpha-catenin trailing myosin (and not beta-catenin). The authors either need to add more data to Figure 1 or revise the text to accurately describe the more limited figure.

2. The text at the top of page 15 (rest of a paragraph) could mostly be moved to the methods.

3. Should the authors refer to the “internalization” of the analyzed cells, rather than to “gastrulation” given that the cells undergoing internalization are not endodermal?

4. The authors use the term “master” regulator. Like chromosome “segregation” during mitosis, these terms (even more so, “master” regulator) reek of structural racism and should be disposed of in the scientific literature. The concept of a master regulator in particular has been absurd for decades now, given the complexity of biology. The use of the term seems more geared to promoting the reputation of investigators than to advancing our understanding of biology in any meaningful way. Proteins are required or not. Let’s drop the sensationalism. And especially please stop using that offensive and absurdly ego-oriented term, master-regulator. It makes me not want to read this paper again.

Reviewer #2: Slabodnick et al describe two screens (proteome and transcriptome) for regulators of C. elegans gastrulation, and characterize the function of two of the hits, namely afadin and zyxin, to couple the contracting apical actomyosin to cell junctions to exert the pulling force. While C. elegans is renowned as a genetic system, the gastrulation process has long defied genetic screens. This study continues a long-term and admirable effort of the Goldstein lab to push the screens with cutting edge techniques, and the results are valuable to the field. The fast live imaging and the semi-automatic image analysis provide solid support for the function of afadin and zyxin to couple actomyosin and cell junctions. My comments are on minor details of the manuscript.

1. Table 1 names the developmental stages as 8- and 16-cell, while the main text as 8- and 24-cell. Please reconcile.

2. Fig 1B. It appears that the conclusion comes from the kymograph, and the line scan does not seem to contribute. If I missed the point, please clarify. Otherwise, consider removing or de-emphasizing the line scan panels.

3. Localization of afadin. Fig 2E-F focus on the junction of Ea/p. The more interesting junctions would be those of Ea/p with their surrounding cells that cover Ea/p upon gastrulation. I understand that those junctions move and may be at an awkward angle to image, so it may not be practical to quantify. In that case, please comment/clarify.

Related, the enrichment of afadin at the Ea/p junction appears to be consistent with notion of afadin/canoe responding to tension to secure adhesion (eg, Yu and Zallen, 2020). A brief discussion would help readers to make the connection and have a better understanding of what afadin may do in terms of coupling the mechanical forces.

4. The Gad phenotype. The manuscript states the Gad phenotype as "the two EPCs failing to fully internalize or failing to stay internalized at the time the EPCs divided". These may be two different phenotypes with different interpretations of gene function. In particular, the latter may imply adhesion between the surrounding cells that cover Ea/p. For example, previous studies (eg, Pohl and Bao, 2012) showed dynamic interactions between the surrounding cells to seal over Ea/p. If the live imaging data allow, the authors should try to distinguish between the two phenotypes. If not, it'll be beneficial for the readers to provide a brief clarification/discussion.

5. The transcriptome screen. The scRNA-seq dataset from the Murray and Waterston groups (Packer et al 2019) seems to provide better temporal and lineal resolution in terms of the gastrulating cells outside the E lineage. Could the authors comment on why they chose to conduct a marker-based sequencing?

6. The concept of slippage between actomyosin and cell boundary is central to the conclusions on afadin and zyxin function. It may be worthwhile to move the one-sentence definition of slippage from Methods to the main text to help the readers.

Reviewer #3: The paper by Slabodnick, Tintori et. al. attempts to shed light on the mechanism by which the contracting apical actomyosin network is coupled to cell-cell junctions in order to facilitate apical constriction and cell invagination, using C. elegans gastrulation as a model. The paper sets out to find new proteins that could either serve as physical linkers themselves or regulate the timing of engagement of such linkers, both in the endodermal precursor cells and more widely, in other gastrulating cells. The paper describes a mass spec experiment to identify proteins co-immunoprecipitating with HMP-1/α-catenin, data mining of published RNA-seq experiments and new single-cell RNAseq experiments to identify transcripts enriched specifically in ingressing cells just prior to their ingression. Based on these, the authors choose 32 candidate genes with which they perform an RNAi by injection screen for the Gad phenotype in EPCs. Of the resulting hits, they choose to follow up with some experiments that address the involvement of AFD-1 and ZYX-1 in actomyosin-junction coupling. The paper shows that systemic RNAi of afd-1 results in 25% gastrulation defects of Ea/Ep, that AFD-1 co-localizes with the cadherin/catenin complex, and that its junctional localization is cadherin-dependent. The paper also shows that systemic RNAi of zyx-1 results in 25% gastrulation defects of Ea/Ep and a zyx-1 null mutant has 58.8% Gad phenotype. RNAseq showed that zyx-1 transcripts are enriched the E lineage, and a highly overexpressed ZYX-1 fluorescent fusion protein could be detected in small foci at the apical junction as well as in the cytoplasm, but fluorescence microscopy failed to show expression of endogenously tagged ZYX-1. The paper describes a new semi-automated image analysis pipeline to analyze centripetal myosin flows, membrane movement, and the correlation between them in order to calculate slippage during Ea/Ep apical constriction, and they use this tool to show that slippage reduction over time is attenuated in zyx-1 null and afd-1 RNAi embryos, suggesting a role for worm zyxin and afadin in linking cell-cell junctions with apical actomyosin during gastrulation.

The paper addresses an important question and pursues logical hypotheses. Experiments are well-controlled and the resulting data are properly analyzed and presented. However, the paper in its current form suffers from three major problems. The first, is that it includes experiments (primarily with negative results) that don’t advance the aims of the paper and don’t contribute to its conclusions. The second, is that the evidence for the conclusion that zyxin and afadin directly link junctions to actomyosin within ingressing cells is weak, and is mostly based on what is known about their function in other systems, which leads me to the third point. Third, in choosing to focus on zyx-1 and afd-1, which are well known cytoskeletal linkers shown to be involved in apical constriction in other systems, and by performing only a limited number of experiments with these proteins, the paper falls short of providing new information on the mechanism of actomyosin-junction coupling.

In my view, in order for the paper to provide sufficient advance to the field to warrant publication in Plos Genetics it should either validate and substantiate the discovery of novel proteins that regulate actomyosin-junction coupling by performing protein localization and imaging of slippage for the other proteins in their list that gave a Gad phenotype, or provide novel mechanistic insight into the function of AFD-1 or ZYX-1 by performing, for example, cell-specific rescue experiments with various deletion constructs in the null background, or by performing more advanced imaging to visualize and quantify junctional and cytoskeletal protein dynamics under control and loss of function conditions.

Specific comments:

1. In search of the link between junctions and apical actomyosin the authors screened for proteins from early-stage embryos that co-immunoprecipitate with α-catenin. This approach assumes that the connection with cadherin is mediated through α-catenin, which may be the case, but is not by any means the only way to connect actomyosin to the cell-cell junctions. The authors should state this limitation in their approach. Furthermore, in order to solubilize intact CCCs the lysis conditions must be fairly harsh, which means many weaker protein interactions would be lost. A BioID approach is probably more appropriate to answer this question.

2. The authors narrow the list of candidates emerging from the Co-IP based on expression before or during the 24-cell stage of development using published single-cell mRNA sequencing data. Why don’t they use the expression data they generated here for other lineages?

3. AFD-1 and ZYX-1 were both candidates based on the EPC expression enrichment dataset. Hence, if I understood correctly, none of the HMP-1 co-IP proteins ended up being pursued on the basis of the co-IP alone. This makes me ask: if this co-IP was not useful why include it in the paper?

4. The results of the RNA-seq of internalizing cell types provides the community with new RNA expression data, but it does not, in the end, advance the goal of this paper. No “master regulator” of apical constriction was found and the idea that a family of LIM domain proteins, which are enriched in various gastrulating lineages, are common regulators of gastrulation is also not supported because RNAi of pxl-1 and lim-9 gave 0% gad. Therefore, in my opinion, this data unnecessarily complicates the paper and should be published separately.

5. Figure 5 and the results section describing the semi-automated workflow for analyzing actomyosin-membrane coupling, which is straight forward and based on established tools, belongs in the materials and methods section and not in the results section.

6. The authors write that they follow up on zyx-1 and afd-1, which gave only 25% gad phenotype, because they are known junctional components. This is puzzling to me because in their RNAi screen several genes that are not known to regulate junction to medioapical actin connection had a higher parentage of gastrulation defects and would be, in my opinion, more interesting to study. For example, gei-4 had 42% Gad, hum-8 had 67%, and 46E10.8 had 50% Gad. As the authors note, afadin/canoe was already shown in Drosophila to be important for connecting medioapical actomyosin to cell-cell junctions, so showing it has a similar role in worms serves as a validation and it not novel.

7. A major caveat with all the RNAi experiments is that they are systemic and not cell specific. This is especially a problem with zyx-1 because it is expressed at very low levels in the Ea/Ep cells and therefore it raises the question whether the RNAi-induced phenotype isn’t due to its depletion in other cells. EPC specific depletion of zyx-1 and afd-1 would be necessary to convincingly show that their expression in these cells is important for their invagination.

8. Not being able to detect endogenous ZYX-1 in the EPCs is also a problem. It is possible to tag endogenous proteins with multiple fluorescent protein copies (e.g. 3XGFP) and this has previously been shown to allow the visualization of weakly expressed proteins. Visualizing endogenous zyxin would allow the authors to follow how its localization changes between early actomyosin contractions and coupled actomyosin contractions.

9. An important question that was not addressed in the paper is how do CCC and actomyosin dynamics change following afd-1 or zyx-1 depletion. The authors used a membrane marker to image junctions and measure slippage, but it might be necessary to image components of the cadherin-catenin complex for this purpose.

10. The paper shows that AFD-1 co-localizes with HMR-1 and that it accumulates at the apical region during apical constriction. However, a more important question that should be addressed is how does its localization or amount change over time between high slippage and low slippage regimes?

Minor comments:

1. In Table 1 the proteins are listed according to alphabetic order of gene name, which makes no sense. They should be divided according to the screen they came out of and ordered according to the severity of their gad phenotype.

2. The authors write at the end of the first results section “these results imply that a strong connection between actomyosin and junctional α-catenin is initially missing in this system.”- It is very likely that there are separate actomyosin networks at cell-cell junctions and in the apical region of the cell. Therefore, it should be stated more clearly that the authors conclude that a connection between the contracting apical actomyosin network and the CCC is initially missing, because there likely exists a connection between HMP-1 and junctional actomyosin at the same stage.

3. The authors found lower myosin and slippage rates overall than previously. It wasn’t clear why this was the case and which numbers are more reflective of reality.

4. Figure 3 should include images of the hmr-1 and sax-7 RNAi embryos

Suggestions for experiments that could add mechanistic insight:

1. The afd-1(RNAi) Gad phenotype is weak and the authors speculate that AFD-1 has a partially redundant role in C. elegans gastrulation. The authors could use an afd-1 mutant as a sensitized background to screen the other candidates by RNAi or do double RNAi with afd-1 plus each of the other candidates.

2. The authors speculate that AFD-1 might be post-translationally regulated differently in EPCs than in other cells. This could be tested with rescue experiments using mutants affecting specific PTMs.

**Have all data underlying the figures and results presented in the manuscript been provided?**

Reviewer #1: Yes

Reviewer #2: **No: **Availability of proteomics data is not described.

Reviewer #3: Yes

PLOS authors have the option to publish the peer review history of their article (what does this mean?). If published, this will include your full peer review and any attached files.

Reviewer #1: No

Reviewer #2: No

Reviewer #3: No

---

## [Decision Letter · Decision Letter 1]

23 Feb 2023

Dear Dr Slabodnick,

We are pleased to inform you that your manuscript entitled "Zyxin contributes to coupling between cell junctions and contractile actomyosin networks during apical constriction" has been editorially accepted for publication in PLOS Genetics. Congratulations!

Yours sincerely,

Andrew D. Chisholm

Academic Editor

PLOS Genetics

Gregory Barsh

Editor-in-Chief

PLOS Genetics

Comments from the reviewers (if applicable):

Reviewer's Responses to Questions

**Comments to the Authors:**

Reviewer #1: The authors have done an admirable job of addressing the comments from all three reviewers, and have added new data, and importantly clarified their emphasis on "dynamic coupling" as a key advance. In my opinion, the authors have thoroughly and sufficiently addressed all reviewer comments and have substantially improved their manuscript. The advances reported warrant publication in PLOS genetics with no further revisions needed.

Reviewer #2: The authors addressed most of the issues raised by the reviewers. I don't have further comments

**Have all data underlying the figures and results presented in the manuscript been provided?**

Reviewer #1: Yes

Reviewer #2: Yes

PLOS authors have the option to publish the peer review history of their article (what does this mean?). If published, this will include your full peer review and any attached files.

Reviewer #1: **Yes: **Bruce Bowerman

Reviewer #2: No

**Data Deposition**

http://datadryad.org/submit?journalID=pgenetics&manu=PGENETICS-D-22-00784R1

**Press Queries**

---

## [Editor Report · Acceptance letter]

23 Mar 2023

PGENETICS-D-22-00784R1 

Zyxin contributes to coupling between cell junctions and contractile actomyosin networks during apical constriction 

Dear Dr Slabodnick, 

We are pleased to inform you that your manuscript entitled "Zyxin contributes to coupling between cell junctions and contractile actomyosin networks during apical constriction" has been formally accepted for publication in PLOS Genetics! Your manuscript is now with our production department and you will be notified of the publication date in due course.

With kind regards,

Anita Estes

PLOS Genetics

On behalf of:
